# LoRA Learns Less and Forgets Less

**Dan Biderman**[1,2], **Jacob Portes**[2], **Jose Javier Gonzalez Ortiz**[2], **Mansheej Paul**[2], **Philip Greengard**[1], **Connor Jennings**[2], **Daniel King**[2], **Sam Havens**[2], **Vitaliy Chiley**[2], **Jonathan Frankle**[2], **Cody Blakeney**[2], **John P. Cunningham**[1]

[1]**Columbia University** {db3236, pg2118, jpc2181}@columbia.edu
[2]**Databricks Mosaic Research** {jacob.portes, j.gonzalez, mansheej.paul, connor.jennings, daniel.king, sam.havens, vitaliy.chiley, jfrankle, cody.blakeney}@databricks.com

Reviewed on OpenReview: https://openreview.net/forum?id=aloEru2qCG

## Abstract

Low-Rank Adaptation (LoRA) is a widely-used parameter-efficient finetuning method for large language models. LoRA saves memory by training only low rank perturbations to selected weight matrices. In this work, we compare the performance of LoRA and full finetuning on two target domains, programming and mathematics. We consider both the instruction finetuning ($\approx$100K prompt-response pairs) and continued pretraining ($\approx$20B unstructured tokens) data regimes. Our results show that, in the standard low-rank settings, LoRA substantially underperforms full finetuning. Nevertheless, LoRA better maintains the base model's performance on tasks outside the target domain. We show that LoRA mitigates forgetting more than common regularization techniques such as weight decay and dropout; it also helps maintain more diverse generations. Finally, we show that full finetuning learns perturbations with a rank that is 10-100$\times$ greater than typical LoRA configurations, possibly explaining some of the reported gaps. We conclude by proposing best practices for finetuning with LoRA.

## 1 Introduction

Finetuning large language models (LLMs) with billions of weights requires a non-trivial amount of GPU memory. Parameter-efficient finetuning methods reduce the memory footprint during training by freezing a pretrained LLM and only training a small number of additional parameters, often called adapters. Low-Rank Adaptation (LoRA; Hu et al. (2021)) trains adapters that are low-rank perturbations to selected weight matrices.

LoRA is widely adopted for finetuning LLMs under hardware constraints, but the jury is still out on whether it compromises performance compared to full finetuning. The two seminal methods papers on the topic, introducing LoRA (Hu et al., 2021) and its more recent combination with model quantization (QLoRA; Dettmers et al. (2024)) reported that LoRA performs better or equivalent to full finetuning. More empirical work (Ghosh et al., 2024; Zhao et al., 2024b) reaches a similar conclusion; and this sentiment is echoed in an array of industry blog posts as well (e.g., Raschka (2023); Niederfahrenhorst et al. (2023)). At the same time, there is evidence that LoRA underperforms full finetuning (Ivison et al., 2023; Zhuo et al., 2024), and the need to improve upon LoRA has led to the development of enhanced LoRA variants (Hayou et al., 2024; Meng et al., 2024; Li et al., 2023b; Shi et al., 2024) or alternative low-rank approximation methods (e.g Liu et al. (2024); Zhao et al. (2024a)). To shed light on this ongoing debate, **we ask: under which conditions does LoRA approximate full finetuning accuracy on challenging target domains, such as code and math?**

By training fewer parameters, LoRA is hypothesized to constrain the finetuned model from diverging significantly from the base model (Sun et al., 2023; Du et al., 2024). This potential characteristic is particularly helpful for LLM finetuning, a form of continual learning where specializing in new domains can

come at the expense of base model capabilities (Wang et al., 2024) (a phenomenon known its extreme form as "catastrophic forgetting" McCloskey & Cohen (1989); French (1999)). To date, only a few studies have examined forgetting in modern LLMs (Kleiman et al., 2023; Kalajdzievski, 2024; Vu et al., 2022). To address this gap, **we also ask: when performing continual learning on a new domain, to what extent does LoRA mitigate forgetting of base model capabilities?**

In this study, we compare LoRA and full finetuning for Llama-2 7B models across two challenging target domains, code and mathematics. Within each domain, we explore two training regimes. The first regime is *continued pretraining*, which involves training on billions of unlabeled domain-specific tokens, most commonly via full finetuning; here we use the StarCoder-Python (Li et al., 2023a) and OpenWebMath (Paster et al., 2023) datasets (Table 1). The second is *instruction finetuning*, the common scenario for LoRA involving question-answer datasets with tens to hundreds of millions of tokens. Here, we use Magicoder-Evol-Instruct-110K (Wei et al., 2023) and MetaMathQA (Yu et al., 2023).

We evaluate target-domain performance (henceforth, *learning*) via challenging coding and math benchmarks (HumanEval; Chen et al. (2021), and GSM8K; Cobbe et al. (2021)). We evaluate source-domain *forgetting* performance on language understanding, world knowledge, and common-sense reasoning tasks (Zellers et al., 2019; Sakaguchi et al., 2019; Clark et al., 2018).

We find that with commonly used low-rank settings, LoRA substantially underperforms full finetuning, while typically requiring longer training (Sec. 4.1). In continued pretraining, the performance gap between full finetuning and LoRA is not closed even with high ranks. In instruction finetuning, on the other hand, high ranks can match full finetuning performance.

Despite LoRA's limitations, we show that it consistently maintains better source-domain performance compared to full finetuning (Sec. 4.2). Furthermore, we characterize the tradeoff between learning and forgetting (Sec. 4.3). We then show that LoRA – even with higher rank – mitigates forgetting more aggressively than classic regularization techniques that aim to prevent overfitting, such as dropout (Srivastava et al., 2014; Goodfellow et al., 2013), and weight decay (Goodfellow et al., 2016). Moreover, by analyzing the generated solutions to HumanEval problems, we demonstrate that while full finetuning tends to produce a limited set of solutions, LoRA produces a wider range of solutions more akin to those of the base model (Sun et al., 2023; Du et al., 2024)

Why does LoRA underperform full finetuning? LoRA was originally motivated in part by the hypothesis that finetuning results in low-rank perturbations to the base model's weight matrix (Li et al., 2018; Aghajanyan et al., 2020; Hu et al., 2021). However, the tasks explored by these prior works are relatively easy for modern LLMs, and certainly easier than the coding and math domains studied here. Thus, we perform a singular value decomposition to show that full finetuning barely changes the spectrum of the base model's weight matrices, and yet the difference between the two (i.e. the perturbation) is high rank. The rank of the perturbation grows as training progresses, with ranks 10-100× higher than typical LoRA configurations (Figure 6).

We conclude by proposing best practices for training models with LoRA. We find that LoRA is very sensitive to hyperparameters, including learning rates, choice of target modules, ranks, and scaling factors; setting these properly is a prerequisite to approach full finetuning performance.

To summarize, we contribute the following results:

- Full finetuning is more accurate and sample-efficient than LoRA in CPT for code and math; in instruction finetuning, higher ranks can close most of the gaps (Sec.4.1).

- LoRA forgets less of the source domain (Sec. 4.2 and 4.3).

- LoRA forgets less than common regularization techniques; it also helps maintaining the diversity of generations (Sec. 4.5).

- Full finetuning finds high rank weight perturbations (Sec. 4.6).

- A hyperparameter sensitivity analysis for LoRA, as well as practical recommendations (Sec. 4.7).

Model checkpoints and LoRA adapters can be accessed at `https://github.com/danbider/lora-tradeoffs`.

| | Code | Math |
|---|---|---|
| CPT | StarCoder-Python (up to 20B tokens) | OpenWebMath (up to 20B tokens) |
| IFT | Magicoder-Evol-Instruct-110K (72.3M tokens) | MetaMathQA (103M tokens) |

Table 1: Datasets and token counts for math and code experiments

## 2 Background

LoRA involves freezing a pretrained weight matrix $W_{\text{pretrained}} \in \mathbb{R}^{d \times k}$, and learning only a low-rank perturbation to it, denoted here as $\Delta$, as follows:

$$W_{\text{finetuned}} = W_{\text{pretrained}} + \Delta$$
$$\Delta = \gamma_r AB, \quad A \in \mathbb{R}^{d \times r}, \quad B \in \mathbb{R}^{r \times k}.$$

Most common implementations initialize $A_0 \sim \mathcal{N}(0, 1)$, $B_0 = 0$ and set the scalar $\gamma_r = \alpha/r$ with a controllable hyperparameter $\alpha$. The user chooses which $W_{\text{pretrained}}$ to adapt ("target modules"), the rank $r << d, k$, and the hyperparameter $\alpha$. By doing so, only $d \times r + r \times k$ parameters are trained per module instead of $d \times k$, which reduces the memory and FLOPS required for computing the gradient. As an example, applying a $r = 16$ LoRA to a 7B weight matrix with $d = k = 4096$ trains $< 1\%$ of the original parameter count. Appendix Sec. H lays out the approximate memory savings by LoRA.

LoRA's introduction and first applications targeted only the $W_q$ and $W_v$ matrices in the self-attention module (Hu et al., 2021). Since then, it has become best practice to target all transformer modules (Raschka, 2023; Dettmers et al., 2024), i.e., $\{W_q^{(l)}, W_k^{(l)}, W_v^{(l)}, W_o^{(l)}\}_{l=1}^L$ in the self-attention modules, and $\{W_{\text{gate}}^{(l)}, W_{\text{up}}^{(l)}, W_{\text{down}}^{(l)}\}\}_{l=1}^L$ in the feedforward modules for $L$ layers in, say, a Llama architecture (Hu et al., 2021; Touvron et al., 2023).

## 3 Experimental Setup

We train on code and math datasets that have been shown to increase downstream performance. We motivate the training datasets and evaluation benchmarks below. All training was done using the Databricks MosaicML `composer`[1], `streaming`[2], and `llm-foundry`[3] repositories, as well as the HuggingFace `peft` library.

### 3.1 Datasets for Continued Pretraining (CPT) and Instruction Finetuning (IFT)

**Coding CPT - Starcoder-Python** (Li et al., 2023a) This dataset consists of permissively licensed repositories from GitHub, including Git commits, in 80+ programming languages. We chose the Python subset and sub-sampled it to 20B tokens.

**Math CPT - OpenWebMath** (Paster et al., 2023) This dataset contains 14.7B tokens derived from mathematical web pages from Common Crawl, correctly formatted to preserve mathematical content such as LaTeX equations.[4] To match with the StarCoder-Python dataset, we trained on up to 20B tokens, repeating tokens beyond the first 14.7B. An analysis of this dataset shows that it contains a considerable amount of full English sentences.[5]

**Coding IFT - Magicoder-Evol-Instruct-110k** (Wei et al., 2023) This dataset contains 72.97M tokens of programming questions and answers. It reproduces the "Evol-Instruct" dataset of WizardCoder (Luo et al.,

---

[1] `https://github.com/mosaicml/composer`

[2] `https://github.com/mosaicml/streaming`

[3] `https://github.com/mosaicml/llm-foundry`

[4] `https://huggingface.co/datasets/open-web-math/open-web-math`

[5] Out of a random selection of 100K examples, a regex search shows that 75% of the examples contain LaTex. The data is classified as 99.7% English and "overwhelmingly English" by the `langdetect` and `fasttext` tools.

2023b) by iteratively prompting an LLM (GPT-4) to increase the difficulty of a set of question-answer pairs from Code Alpaca (Chaudhary, 2023).

**Math IFT - MetaMathQA** (Yu et al., 2023) This dataset was built by bootstrapping mathematical word problems from the *training* sets of GSM8K (Cobbe et al., 2021) and MATH (Hendrycks et al., 2021) by rewriting the questions with variations using GPT-3.5. This dataset contains 395K question-answer pairs and roughly 103M tokens.[6]

We quantify learning and forgetting via benchmarks reported on the Open LLM Leaderboard[7] for state of the art open-source LLMs such as Llama (Touvron et al., 2023).

### 3.2 Measuring Learning with Coding and Math Benchmarks (*target domain* evaluation)

**Coding - HumanEval** (Chen et al., 2021) This benchmark contains 164 problems that involve generating of a Python program given a docstring and a function signature. A generation is considered correct if it passes all supplied unit tests. We use the Code Generation LM Evaluation Harness (Ben Allal et al., 2022) configured to output 50 generations per problem, and calculate "pass@1" with softmax temperature=0.2 and $top\_p = 0.95$ for 0-shot HumanEval.

**Math - GSM8K** (Cobbe et al., 2021) This benchmark includes a collection of 8.5K grade-school math word problems. We evaluate on the test split of GSM8K (1,319 samples) as implemented in the LM Evaluation Harness (Gao et al., 2023), with default generation parameters (temperature=0, 5 few-shot, pass@1).

### 3.3 Forgetting Metrics (*source domain* evaluation)

We use the following benchmarks to asses degradation of base model capabilities. **HellaSwag** (Zellers et al., 2019) includes 70K problems, each describing an event with multiple possible continuations. The task is to pick the most plausible continuation, which requires making inferences about nuanced everyday situations. **WinoGrande** (Sakaguchi et al., 2019) also assesses commonsense reasoning. It includes 44K problems with sentences that require ambiguous pronoun resolution. **ARC-Challenge** (Clark et al., 2018) consists of 7,787 grade-school level, multiple-choice science questions, and tests complex reasoning and understanding of scientific concepts. These benchmarks involve multiple-choice questions that use the predicted logits for calculating accuracy, and do not require specifying further generation hyperparameters. All forgetting metrics were computed using the MosaicML Gauntlet evaluation harness (Dohmann, 2023).[8]

## 4 Results

### 4.1 Target-domain performance: LoRA at low ranks underperforms full finetuning

We compare LoRA and full finetuning after performing an exhaustive learning rate sweep for each method, which we found to be crucial (Dettmers et al., 2024). We include learning rate sweep results in Figure S1.

We perform a sample-efficiency analysis – i.e., compute the learning metrics as a function of training samples seen – for both LoRA and full finetuning. For IFT, we train separate models for 1, 2, 4, 8, and 16 epochs. For CPT, we vary the number of training tokens ($0.25, 0.5, 1, 2, 4, 8, 16, 20$ billion), using individual learning rate cooldown schedules. For each condition, we train one full finetuning model and three LoRA models with ranks $r = 16, 64, 256$ noting that most LoRA papers use a "low" rank of 8-64, (e.g., Dettmers et al. (2024); Zhuo et al. (2024)). The LoRA models target all transformer modules and use $\alpha = 2r$, as known to be best practice (Raschka, 2023). For further details on experimental setup and hyperparameters, see Appendix A.

The results appear in Fig. 1. We first note that for both programming and math, IFT improves evaluation scores much more than CPT, which is expected because the samples in each IFT dataset are more similar to the evaluation problems (e.g., for code, IFT achieves maximum HumanEval of 0.497 vs. 0.263 for CPT).

---

[6]https://huggingface.co/datasets/meta-math/MetaMathQA
[7]https://huggingface.co/spaces/HuggingFaceH4/open_llm_leaderboard
[8]https://github.com/mosaicml/llm-foundry/tree/main/scripts/eval

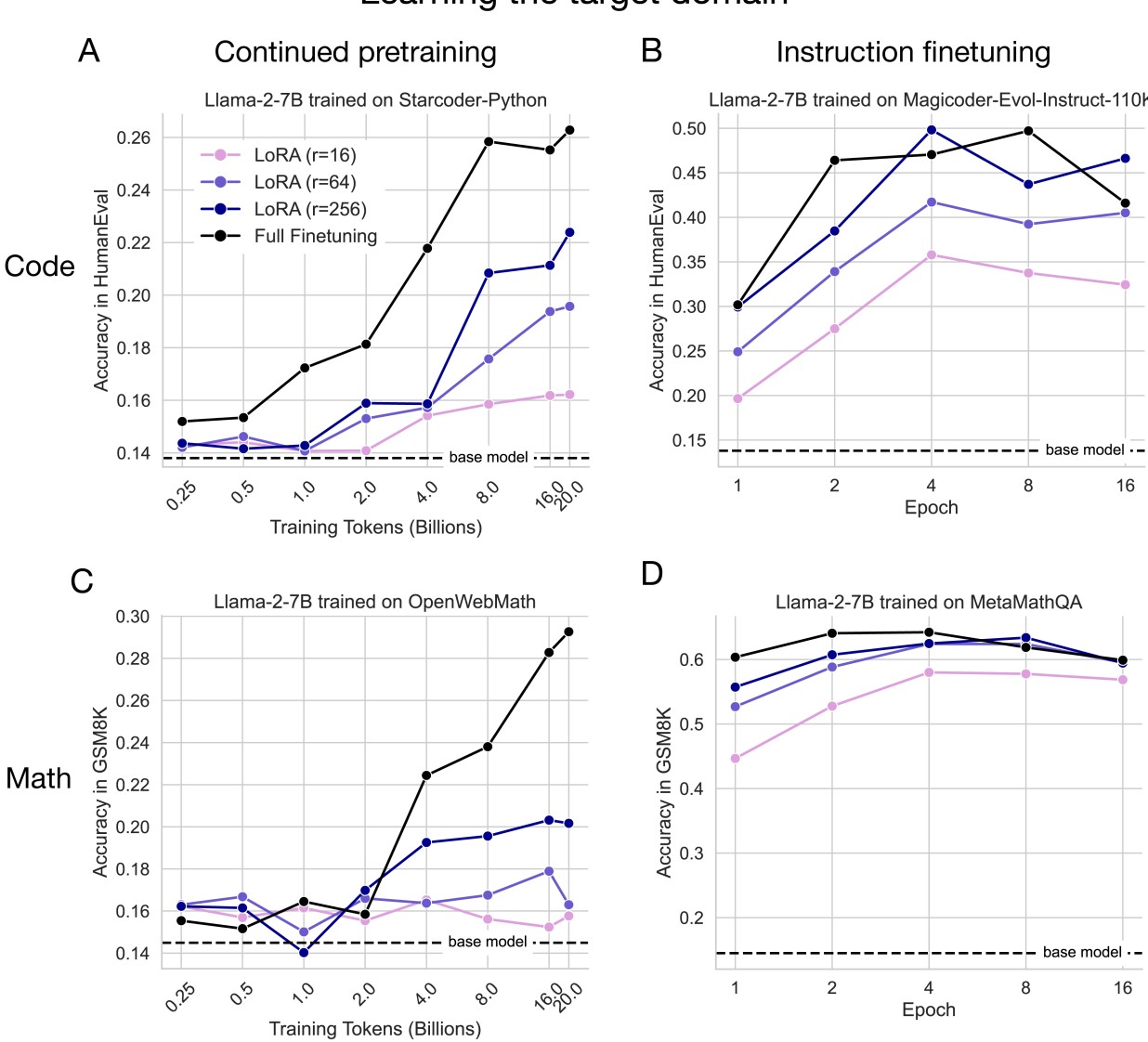

Figure 1: **LoRA performance scales by rank and underperforms full finetuning in code and math.**
(*A*) Starcoder-Python, (*B*) Magicoder-Evol-Instruct-110K, (*C*) OpenWebMath, (*D*) MetaMathQA. In (*A*)
and (*B*) *y*-axis: HumanEval pass@1. In (*C*) and (*D*) *y*-axis: GSM8K strict match. In all panels, "base model"
indicates Llama-2-7b without instruction finetuning. Note that 16 epochs are ≈1.16B and ≈1.6B tokens, for
Magicoder-Evol-Instruct-110K and MetaMathQA.

For **Code CPT** (Fig. 1A and Table S1), we identify a substantial gap between full finetuning and
LoRA that grows with more data. The best LoRA model, with rank $r = 256$, peaks at 20B tokens with
HumanEval=0.224, roughly matching full finetuning with 4B tokens (HumanEval=0.218). Full finetuning
reaches its peak HumanEval of 0.263 at 20B tokens. A clear ordering by rank emerges after the initial 1B
CPT tokens.

For **Code IFT** (Fig. 1B and Table S5), HumanEval accuracy is clearly ordered by rank from the very first
epoch. The more common $r = 16$ and $r = 64$ LoRA configurations have lower accuracy than full finetuning,
with HumanEval scores of 0.358 and 0.417 at epoch 4, respectively). With a high LoRA rank ($r = 256$),
full finetuning performance can be matched (LoRA=0.498 in epoch 4, full finetuning=0.497 in epoch 8).

In Appendix Sec. F we perform a more sensitive HumanEval analysis, calculating pass@$k$ as a function of $k = 1, \dots, 256$ with a higher temperature of 0.8 for full finetuning and the LoRA models (at epoch 4). This analysis shows that full finetuning is superior to $r = 256$ for $k < 64$, after which the two are equal.

**Math CPT** (Fig. 1C and S3) results closely echo those of code CPT. Consistent patterns in GSM8K emerge at 4B tokens. Full finetuning opens a gap in GSM8K which widens with more data. Similarly, LoRA performance is ordered by rank. The best LoRA ($r = 256$) peaks at 16B tokens (GSM8K=0.203), underperforming full finetuning at 4B tokens (GSM8K=0.224) and at its peak at 20B tokens (GSM8K=0.293).

LoRA closes much of the gap with full finetuning in the **Math IFT** (Fig. 1D and Table S7) dataset, while remaining less sample efficient. Both methods substantially improve upon the base model; LoRA ($r = 256$) peaks at 8 epochs (GSM8K=0.634) while full finetuning achieves GSM8K=0.641 at 2 epochs and peaks at 4 epochs, with GSM8K=0.642.[9] Unlike the code IFT dataset, $r = 64$ suffices to approach full finetuning and achieve GSM8K=0.624 at epoch 4. We suggest that lower ranks are effective here because English mathematics problems involve a smaller domain shift from the pretraining data as compared to coding ones.

In summary, in CPT, LoRA underperforms full finetuning across all configurations. In IFT, and especially in code, high LoRA ranks are required to close the gap with full finetuning.

## 4.2 LoRA forgets less than full finetuning

Here, we investigate the extent of forgetting (defined in Sec. 3.2) as a function of training data in Fig. 2.

Overall, we observe that (1) IFT induces more forgetting than than CPT, (2) programming induces more forgetting than math, and (3) forgetting tends to worsen with training duration. Most importantly, LoRA forgets less than full finetuning, and the extent of forgetting is controlled by rank. In code – both CPT and IFT – full finetuning forgets substantially more than any LoRA configuration. In code CPT (Table S2), at 20B tokens, full finetuning scores 0.545 versus 0.617 by LoRA $r = 256$. In code IFT (Table S6), full finetuning scores 0.414 versus 0.509 by LoRA $r = 64$. In math – for both CPT and IFT – LoRA with $r = 256$ forgets nearly as much as full finetuning. In CPT (Table S4), LoRA scores 0.616 (20B tokens) versus 0.613 of full finetuning (16B tokens). In IFT (Table S8), LoRA and full finetuing respectively degrade to 0.567 and 0.559 at epoch 16.

We note that the least forgeting occurs for the OpenWebMath dataset, which is dominated by English sentences (see 3.1 for details).

## 4.3 The Learning-Forgetting Tradeoff

It is trivial that models that change less when finetuned to a new target domain will forget less of the source domain. The nontrivial question is: do LoRA and full finetuning differ in how they tradeoff learning and forgetting? Can LoRA achieve similar target domain performance but with diminished forgetting?

We form learning-forgetting Pareto curves by plotting the forgetting metric versus the learning metric for each training duration (Fig. 3). As models train on more data, they learn more and forget more, traveling up and left in this space. As we increase LoRA ranks, we find that the curves shift up and left as well, again, learning more and forgetting more, doing so more consistently in IFT than CPT.

Each dataset presents a unique tradeoff pattern which makes it difficult to conclude whether LoRA and full finetuning offer fundamentally different learning-forgetting tradeoffs. We will review each dataset next.

For Code CPT, though the full finetuning curve reaches much higher values of HumanEval, it appears to forget more for any given HumanEval value, which LoRA can reach if trained on more tokens. This pattern does not hold for math CPT, where LoRA and full finetuning curves are roughly overlapping until full finetuning shoots up (in 4B tokens) to achieve much higher GSM8K scores without increased forgetting. In code IFT, LoRA $r = 256$ offers comparable HumanEval accuracy while strictly forgetting less. Lower ranks

---

[9]We note that the original MetaMath paper reports a maximum accuracy of 0.665 when (fully) finetuning Llama-2-7B on the MetaMathQA dataset. We attribute this to small differences in hyperparameters; they trained on 3 epochs with a batch size of 128 using the AdamW optimizer, a learning rate of 2e-5, a learning rate warmup of 3%.

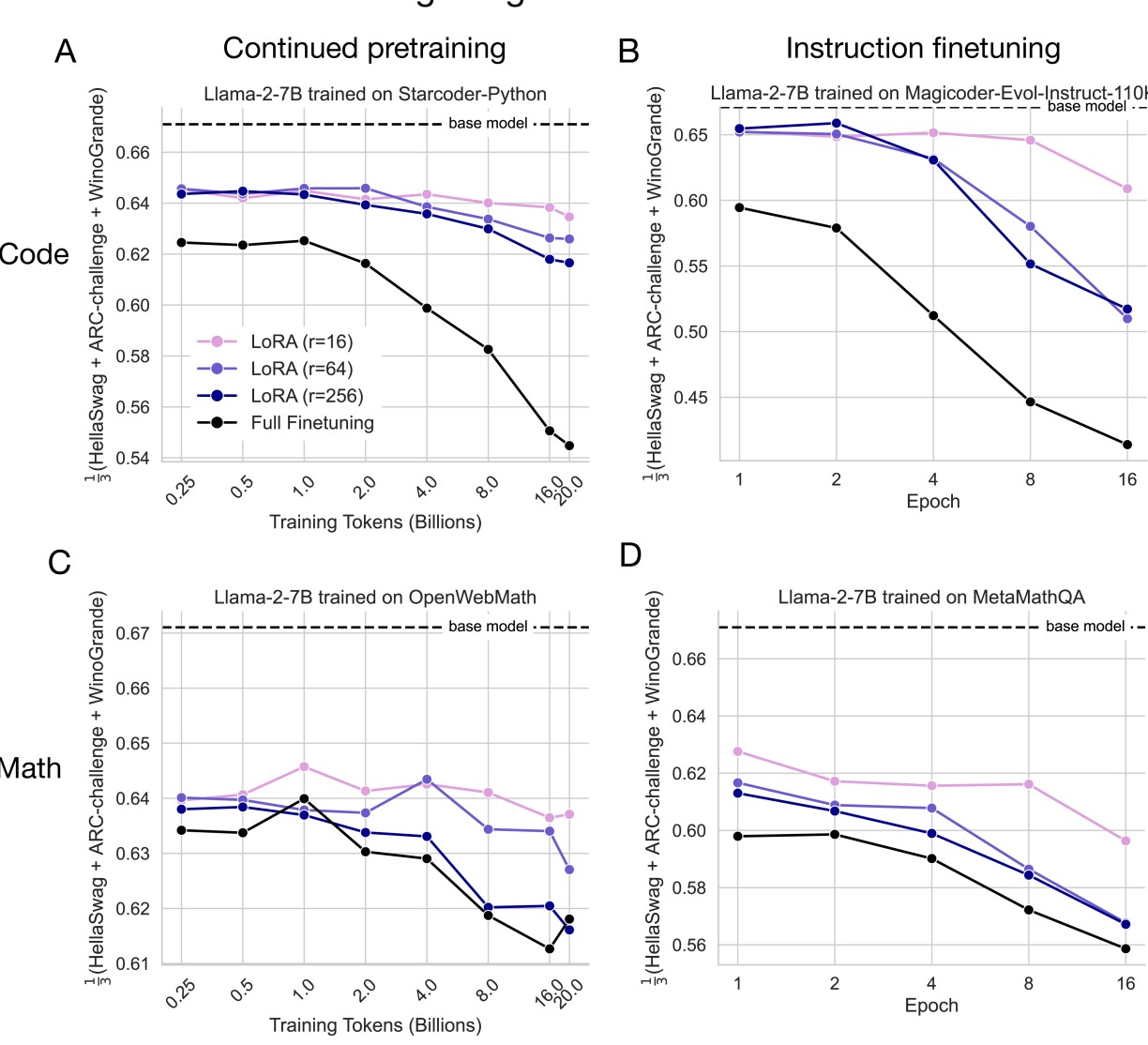

Figure 2: **LoRA forgets less than full finetuning.** In all panels, the *y*-axis shows the average of HellaSwag, ARC-Challenge and Winogrande for Llama-2-7B trained trained on: (A) StarCoder-Python (B) Magicoder-Evol-Instruct-110k (C) OpenWebMath (D) MetaMathQA.

do not reach high values of HumanEval to compare to full finetuning. In math IFT, LoRA and full finetuning seem to lie on adjacent learning-forgetting tradeoff curves, with full finetuning offering preferable tradeoffs.

With the caveats mentioned above, it seems that LoRA can offer preferable learning-forgetting tradeoffs for code, while full finetuning can offer preferable tradeoffs for math. Moreover the choice of LoRA rank can serve as a knob to navigate the learning-forgetting tradeoffs.

### 4.4 For the Tülu-v2-mix dataset, LoRA is on par with full finetuning

So far, we analyzed how LoRA and full finetuning specialize in very specific domains. Often, code or math problems appear as part of larger IFT data mixtures that include multi-turn conversations and a variety of other NLP tasks, such as summarization, etc (e.g., Wei et al. (2021)). We therefore finetuned LoRA and full finetuning models on one such popular dataset, the Tülu-v2-mix (Ivison et al., 2023). The results are

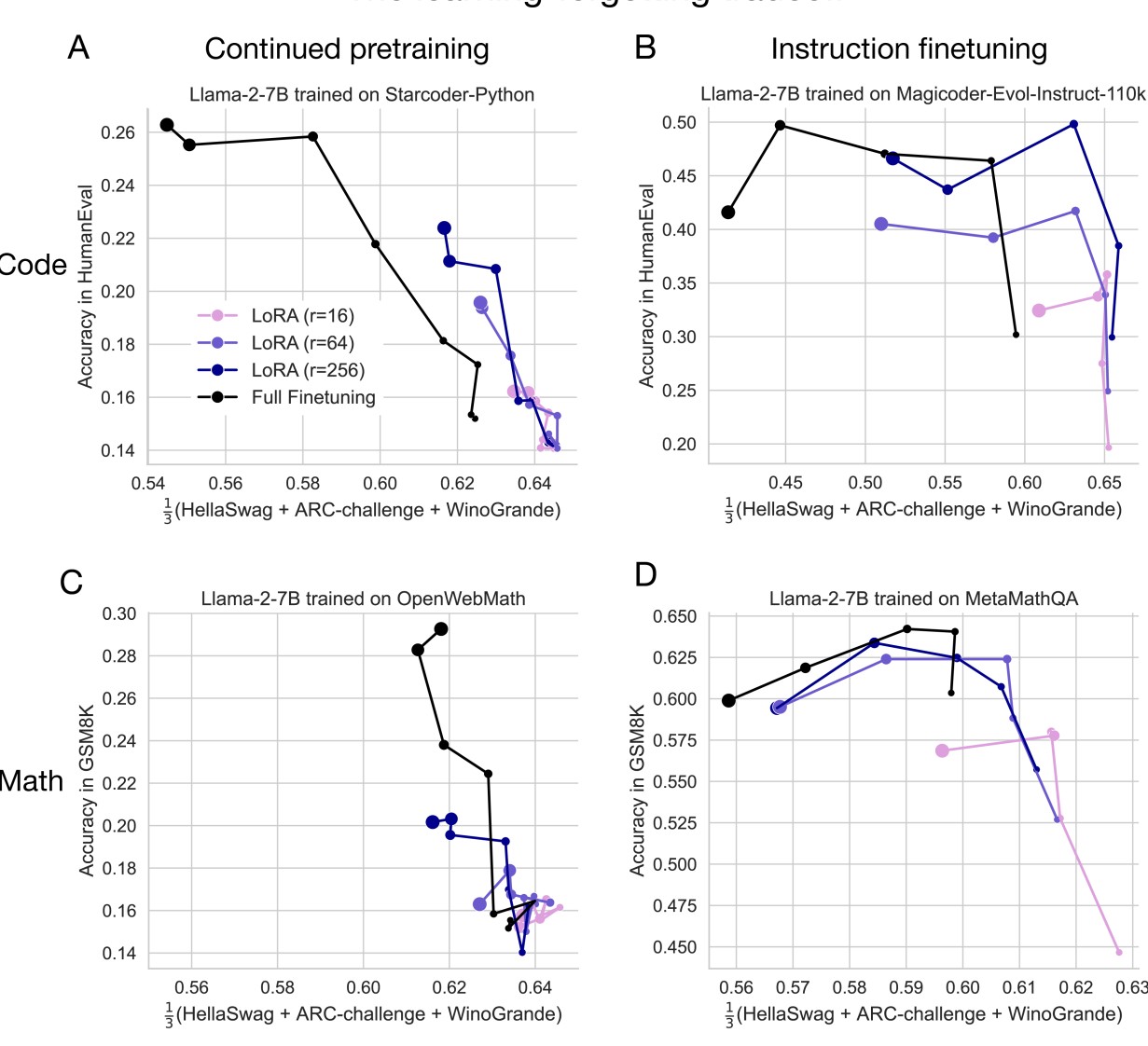

Figure 3: **LoRA vs. full finetuning trade-off for Llama-2-7B.** Relative to full finetuning, LoRA learns less (lower values on the $y$-axis) and forgets less (higher values on the $x$-axis). Each dot is a separate model, with marker size corresponding to training duration (from 0.25-20 billion tokens for CPT, and 1-16 epochs for IFT). Same data as Figures 1, 2.

presented in the Appendix (Sec. C and Table S9). In summary, we find that both LoRA and full finetuning meaningfully improve upon the base model, and that LoRA, even with lower ranks, can match full finetuning in chat quality as measured by Multi-Turn Benchmark (MT-bench (Zheng et al., 2024)), GSM8K (Cobbe et al., 2021), and Massive Multitask Language Understanding (MMLU; Hendrycks et al. (2020)). At longer training durations (6 epochs), LoRA also forgets less.

### 4.5 How strongly does LoRA constrain the finetuning process?

In this section, we analyze Llama-2-7B models trained on the Magicoder-Evol-Instruct-110K dataset. We first compare the learning-forgetting tradeoffs between LoRA and classic regularization techniques, and then analyze the diversity of the generated text.

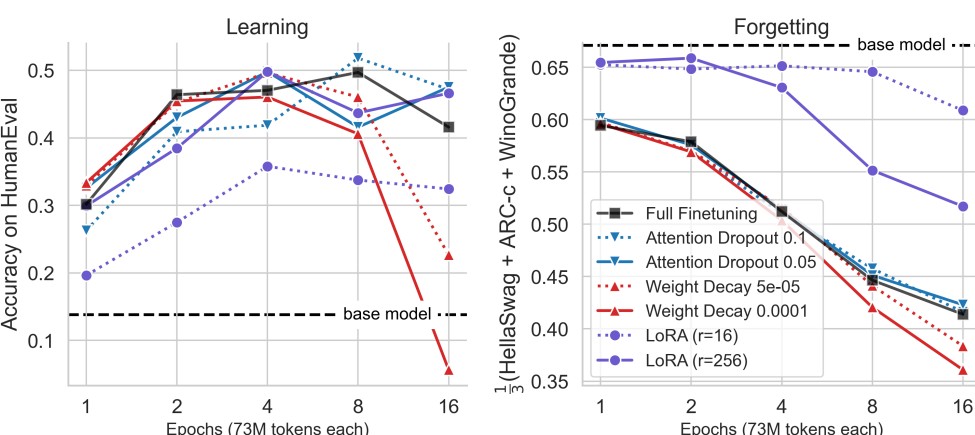

Figure 4: **LoRA forgets less than attention dropout and weight decay.** Results from Llama-2-7B finetuned on Magicoder-Evol-Instruct-110K. Left panel: learning as measured by accuracy on HumanEval. Right panel: forgetting as measured by the average of HellaSwag, ARC-Challenge and WinoGrande scores. The solid slateblue line shows that LoRA (r=256) learns as much as full finetuning, weight decay, and attention dropout, while forgetting much less.

**LoRA forgets less than attention dropout and weight decay** We compare LoRA ($r = 16, 256$, training all modules) to weight decay (Goodfellow et al., 2016) with values $5e^{-5}, 1e^{-4}$ and attention dropout (Srivastava et al., 2014) with values $0.05, 0.1$. Both regularization techniques appear to learn and forget as much as full finetuning, except that weight decay starts to generally deteriorate at longer training durations (epochs 8 and 16). LoRA, with the common $r = 16$, learns less and forgets less than all other models. LoRA $r = 256$, on the other hand, learns as much as the other methods while forgetting less.

**LoRA helps maintain diversity of token generations.** We scrutinize the generated solution strings for HumanEval problems. We calculate the unique number of output strings out of 50 generations (for base model, full finetuning, and LoRA) serving as a coarse proxy for predictive diversity. In Figure 5 we separately show the results for correct and incorrect answers. As in the reinforcement learning from human feedback literature (Du et al., 2024; Sun et al., 2023), we find that full finetuning results in fewer unique generations ("distribution collapse") compared to the base model, for both pass and fail generations, with LoRA in between the two. The above works also suggest that LoRA could even substitute a common Kullback-Leibler divergence term that keeps the probabilities of the generated text similar between the finetuned and base model. We reiterate that exact string matching between generations is not a sensitive metric of predictive diversity, as generations can slightly vary in format and remain functionally identical.

### 4.6 Full finetuning on code and math does not learn low-rank perturbations

In this section, we seek to study whether we should expect low-rank training to be a good approximation to full finetuning, and if so, what is the necessary rank. Recall that full finetuning can be written as $W_{\text{finetuned}} = W_{\text{pretrained}} + \Delta$; here we compute the Singular Value Decomposition of all three terms in the equation. We focus on continued pretraining for code, where there are drastic differences between LoRA and full finetuning. We analyze checkpoints obtained at 0.25, 0.5, 1, 2, 4, 8, 16, and 20 billion training tokens.

First, in Figure S8 we present results for the $W_q$ projection at layer 26 of Llama-2-7B (with dimensions $d \times d$, $d = 4096$). We show that the spectrum of the finetuned weight matrix is very similar to that of the base weight matrix, both decaying slowly and requiring keeping $\approx 50\%$ of singular vectors ($\approx 2000/4096$) to explain 90% of the variance in the weight matrix. Critically, the difference $\Delta$ also has a similar spectrum to the finetuned and base weight matrices (up to a multiplicative scaling). These results are in line with the

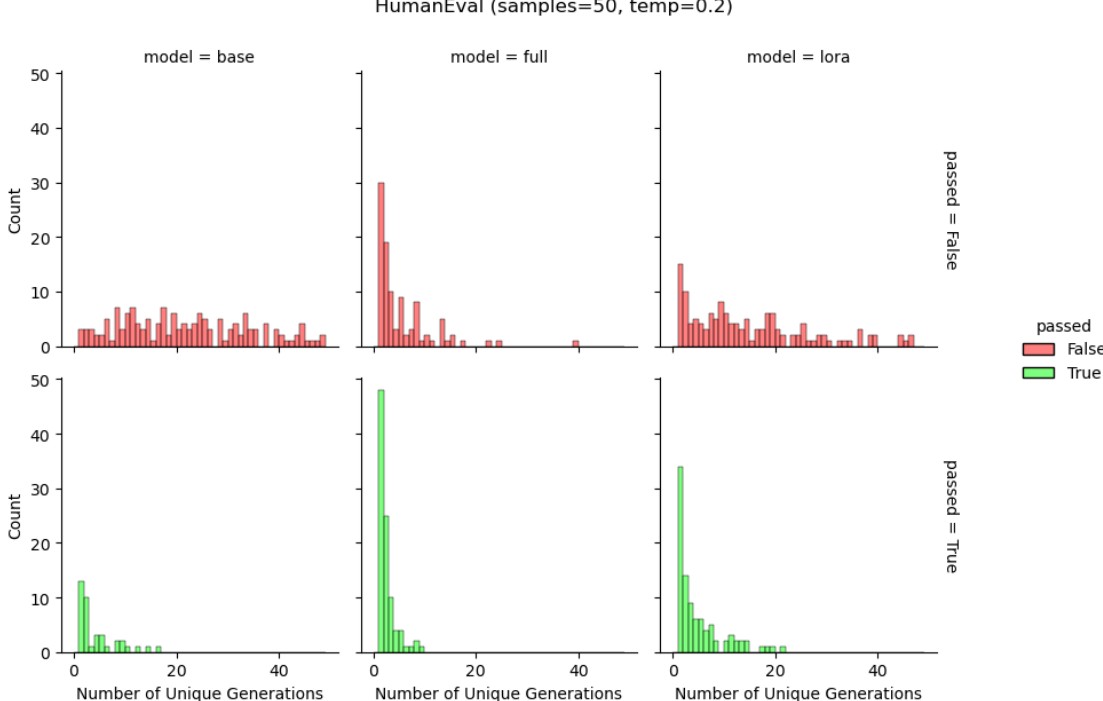

Figure 5: **LoRA maintains output token diversity relative to full finetuning.**

analysis in Zeng & Lee (2024) showing that any transformer model can be well approximated with $r = d/2$. Additionally, we suggest that there is nothing extraordinary about the full finetuning spectra; similar spectra can be achieved by adding low-magnitude Gaussian i.i.d noise to a weight matrix (Fig. S9).

Next, we ask when during training does the perturbation become high rank, and whether it meaningfully varies between module types and layers. We estimate the rank needed to explain 90% of the variance in the matrix. The results appear in Figure 6. We find that: (1) The earliest checkpoint at 0.25B CPT tokens exhibits $\Delta$ matrices with a rank that is $10 - 100\times$ larger than typical LoRA ranks; (2) the rank of $\Delta$ increases when trained on more data; (3) MLP modules have higher ranks compared to attention modules; (4) first and last layers seem to be lower rank compared to middle layers.

### 4.7 Hyperparameter sensitivity analyses for LoRA

Our goal in this work was to optimally configure LoRA so that it has the best chances of matching full finetuning. This is nontrivial, as LoRA has a large number of hyperparameters to choose from: target modules, rank, scaling factors, and learning rates. We turn to analyze the importance of each, and provide some practical recommendations.

First, we found that the choice $\alpha = 2r$ is crucial for high ranks. Most common packages, e.g., HuggingFace's PEFT[10] scale the LoRA matrices by $\alpha/r$, effectively scaling down higher ranks (see also Kalajdzievski (2023)). One might think that high learning rate values may compensate for fixed low $\alpha$'s, but doing so creates instabilities and often leads to inferior performance. To show this, we performed a joint hyperparameter sweep over $\alpha$ and learning rate for the Magicoder dataset training a $r = 256$ LoRA for 4 epochs (Fig. **??**). We find that $\alpha = 512$ does much better than 256 or 32 across all learning rates.

Next, to assess the relative contribution of target modules and rank, we trained Llama-2-7b models on 4 epochs of the Magicoder dataset, sweeping over target modules ("Attention", "MLP", and "All", their union), ranks ($r = 16, 64, 256$), setting $\alpha = 2r$. Fig. 7 shows that HumanEval performance increases with rank, and

---

[10]https://huggingface.co/docs/peft/en/index

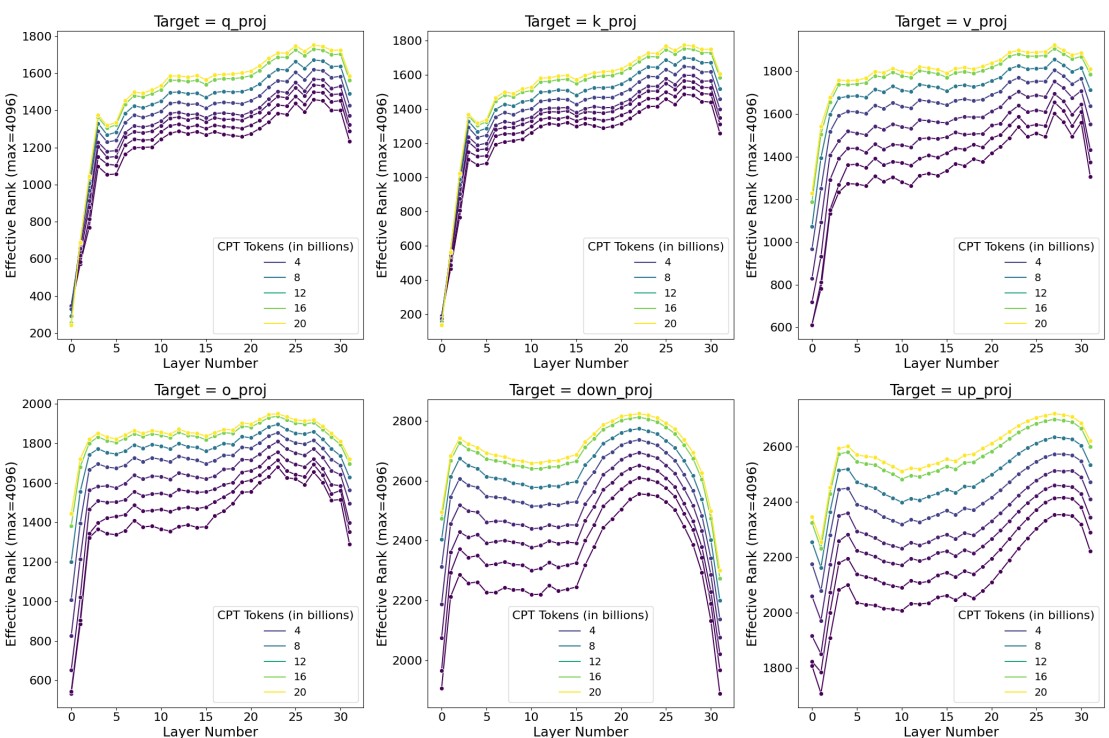

Figure 6: **Dynamics of rank for Llama-2-7B trained on the Starcoder (CPT) data.** In each panel, the x-axis denotes layer number and the y-axis denotes rank needed to explain at least 90% of the variance (maximal dimensionality is 4096). Colors denote CPT tokens, with lighter colors trained for longer.

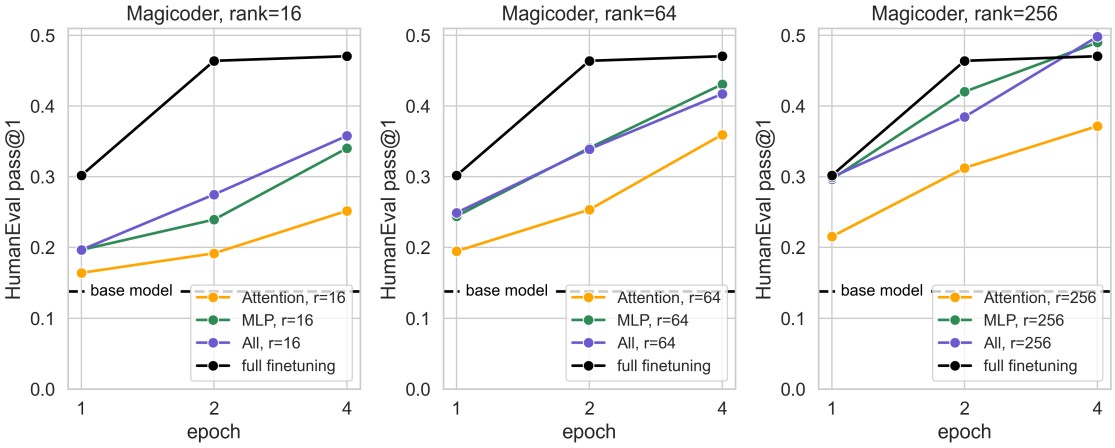

Figure 7: **Targeting `MLP` or `All` modules is superior to training `Attention` modules alone**. All Llama-2-7B checkpoints were trained on Magicoder for 1, 2 and 4 epochs with rank 16 (left), 64 (center) and 256 (right).

that targeting just "Attention" underperforms both "MLP" and "All", where in the latter, most gains are interestingly driven by the "MLP" modules. This is potential evidence that the MLP blocks are the primary loci for continual learning in LoRA, at least in our datasets.

For IFT, we find that LoRA is more sensitive to learning rates compared to full finetuning, and benefits from the highest learning rate that enables stable training for the chosen training duration (see Appendix Sec. B

and Fig. S1). LoRA's best learning rates should be set one order of magnitude higher than full finetuning's, often ranging between $5e^{-5}$ and $5e^{-4}$ for these combinations of model architecture and dataset.

In Appendix Sec. I, we benchmark throughput and peak GPU memory of different LoRA configurations, showing that for standard implementations and a fixed batch size, LoRA tends to train slower than full finetuning.

To conclude, based on our main results and hyperparameter sweeps, we recommend: (a) using LoRA for instruction finetuning and not continued pretraining; (b) if GPU memory allows, targeting "All" transformer modules with a rank of 256, since ranks $16 - 64$ tend not to suffice for code tasks; (c) using $\alpha = 2r$, and (d) sweeping over learning rates between $[1e - 5, 5e - 4]$, picking the highest value that enables stable training.

## 5 Related Work

**Extensions to LoRA**   LoRA has inspired many variants and extensions. One group of methods improves training with LoRA by focusing on initialization or scaling (Meng et al., 2024; Hayou et al., 2024; Li et al., 2023b; Kalajdzievski, 2023; Nikdan et al., 2024), sequential training procedures (Xia et al., 2024), or architectural modifications (Shi et al., 2024). Other works propose alternative low-rank approximations altogether (Liu et al., 2024; Zhao et al., 2024a; Jiang et al., 2024a; Kopiczko et al., 2023). In this study we chose to analyze the classic LoRA setup; while many of these proposed variations of LoRA seem promising, we leave a rigorous comparison of these techniques to future work.

**Benchmarking LoRA vs. Full Finetuning**   The original LoRA paper Hu et al. (2021) reported that LoRA matched full finetuning performance for RoBERTa (Liu et al., 2019) on GLUE (Wang et al., 2018), and GPT-2 on E2E NLG Challenge (Novikova et al., 2017), and GPT-3 on WikiSQL (Zhong et al., 2017), MNLI (Williams et al., 2017), and SAMSum (Gliwa et al., 2019). Many subsequent studies follow this template and report encoder model performance on tasks in GLUE such as SST-2 (Socher et al., 2013) and MNLI (Williams et al., 2017). Models such as RoBERTa are less than 340M parameters, however, and classification tasks such as MNLI are quite trivial for modern billion-parameter LLMs such as Llama-2-7B. Despite LoRA's popularity, only a few studies have rigorously compared LoRA to full finetuning in this setting and with challenging domains such as code and math. Dettmers et al. (2024) for example found that QLoRA matched full finetuning MMLU (Hendrycks et al., 2020) performance when finetuning Llama-1 7B, 13B, 33B and 65B on the Alpaca (Taori et al., 2023) and FLAN (Chung et al., 2024) datasets. Ivison et al. (2023) on the other hand found that QLoRA did not perform as well as full finetuning for Llama-2-7B, 13B and 70B models trained on the Tülü-v2-mix dataset when evaluated across MMLU, GSM8K, AlpacaEval (which uses LLM-as-a-judge; (Dubois et al., 2024)) and HumanEval. One recent notable study is Astraios, which found that LoRA at rank $r = 8$ performed worse than full finetuning on 8 datasets and across 4 model sizes (up to 16 billion parameters), on 5 representative code tasks (Zhuo et al., 2024). Our study corroborates these results and shows that with higher ranks and proper hyperparameter choices, LoRA can perform much better.

The conclusions have also been mixed with regards to the practical details surrounding LoRA target modules and rank: Raschka (2023) and Dettmers et al. (2024) show that optimized LoRA configurations perform as well as full finetuning, and that performance is governed by choice of target modules but *not* rank. However, in that work, the scalar $\alpha$ was not modified with rank, and we found that increasing it to $2r$ was necessary to unlock improvements by rank. In contrast, Liu et al. (2024) shows that LoRA *is* sensitive to ranks. It is likely that some of these discrepancies are due to differences in finetuning datasets and evaluations.

**Continual learning on code and math.**   A growing body of work investigates ways of specializing LLMs to code and math. In code, models such as StarCoder (Li et al., 2023a; Lozhkov et al., 2024), DeepSeek Coder (Guo et al., 2024), and SantaCoder (Allal et al., 2023) were pretrained from scratch on large-scale code datasets. Alternatively, some works start with a generic pretrained base model, and combine continued pretraining on large code datasets followed by IFT on code problems (usually with full finetuning), e.g., Codex (Chen et al., 2021), Code-Qwen (Bai et al., 2023), CodeLlama (Roziere et al., 2023). Some perform only IFT on top of a base model, like MagiCoder Wei et al. (2023), or WizardCoder (Luo et al., 2023b). OctoCoder (Muennighoff et al., 2023) performs IFT with LoRA.

Similarly, much recent work aims to improve mathematical capabilities. Models like DeepSeek Math (Shao et al., 2024) perform continued pretraining on top of a base model, while other methods focus on finetuning by generating high-quality synthetic math problems, scaling to millions of examples. Luo et al. (2023a) takes the Evol-Instruct approach to data generation (akin to the Magicoder dataset; Sec. 3.1) which it then uses to train reward models for instruction quality and solution correctness, which are in turn used for LLM finetuning. Other work develops Monte Carlo Tree Search methods to automatically supervise the intermediate reasoning steps while solving math problems (Luo et al., 2024), and Yue et al. (2024) generates questions and answers from the pretraining web corpus. Toshniwal et al. (2024) uses an LLM to synthesize Code-Interpreter-style solutions to the GSM8K and MATH benchmarks. The proposed solutions can be verified against the official solutions. Singh et al. (2023) iterate over this procedure multiple times ("Self-training") using an expectation-maximization approach. All reviewed methods meaningfully improve math capabilities.

**Learning-Forgetting tradeoffs** Vu et al. (2022) shows that prompt tuning (Lester et al., 2021), another parameter-efficient finetuning method, can aid in mitigating forgetting for cross-lingual summarization tasks (using multilingual variants of the T5 model). With large Llama-style LLMs, it has been reported that code-finetuned LLMs lose some of their capabilities in language understanding and commonsense reasoning (Li et al., 2023a; Roziere et al., 2023; Wei et al., 2023). A common approach to mitigate forgetting involves "replaying" source-domain data during continual learning, which can be done by storing the data in a memory buffer, or generating it on the fly (Lesort et al., 2022; Scialom et al., 2022; Sun et al., 2019).

## 6 Discussion

**Does the difference between LoRA and full finetuning change with model size?** Studies in the past have hinted at a relationship between the effectiveness of finetuning and model size (Aghajanyan et al., 2020; Hu et al., 2021; Zhuo et al., 2024). While recent studies have successfully applied LoRA to 70B parameter models (Ivison et al., 2023; Yu et al., 2023; Niederfahrenhorst et al., 2023; Turgutlu, 2024), and previous work shows that techniques like prompt tuning become more effective for larger models (Vu et al., 2022), we leave a rigorous study of these intriguing scaling properties to future work.

**Limitations of the spectral analysis.** The observation that full finetuning tends to find high rank solutions does not rule out the possibility of low-rank solutions; rather, it shows that they are not typically found. An alternative interpretation is that the rank needed to reconstruct the weight matrix is higher than the rank needed for a downstream task. We also only presented SVD analysis for the continued pretraining setting. It is possible that a similar analysis for the instruction finetuning setting would reveal that the full finetuning does not tend to be as high rank.

## 7 Conclusion

This work sheds light on the downstream performance of 7 billion parameter LLMs trained with LoRA and full finetuning. Unlike most prior work, we use domain-specific datasets in code and math, associated with sensitive evaluation metrics. We show that LoRA, with commonly used low-rank settings, underperforms full finetuning across domains. We also show that LoRA keeps the finetuned model's behavior close to that of the base model, with diminished source-domain forgetting and more diverse generations at inference time. We show that LoRA mitigates forgetting more than classical regularization techniques, and also show that full finetuning finds weight perturbations that are far from being low-rank. We conclude by analyzing LoRA's increased sensitivity to hyperparameters and highlighting best practices.

### Acknowledgements

We would like to thank the editor and the three anonymous reviewers who provided high-quality feedback on this work. We are also grateful to Daniel Han and Damjan Kalajdzievski for carefully reading our work and pointing out the importance of setting $\alpha = 2r$ for training with high ranks.

**Author Contributions**

D.B. led this project by developing code, running experiments, analyzing results, and writing the manuscript. J.P. ran experiments and assisted in the writing of the manuscript. J.G.O. wrote code and ran experiments. P.G. advised the SVD analysis, C.J. ran experiments, and D.K. wrote code. M.P., S.H., V.C., J.F., C.B., and J.P.C. advised this work.

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

## Appendix

## A  Experimental Setup

**LoRA configuration for all experiments**.

All experiments were done with the Databricks MosaicML `composer`, `streaming` and `llm-foundry` libraries in conjunction with the HuggingFace `peft` library on 32×H100-80GB GPUs.

We targeted all trainable modules inside each of the $L$ Llama transformer blocks: $\{W_q^{(l)}, W_k^{(l)}, W_v^{(l)}, W_o^{(l)}, W_{\text{gate}}^{(l)}, W_{\text{up}}^{(l)}, W_{\text{down}}^{(l)}\}\}_{l=1}^{L}$. We used ranks of $r = 16, 64, 256$ and set $\alpha = 2r$, to achieve a constant scaling factor $\gamma_r = 2$ across ranks. We use lora_dropout=0.05.

For both the Code CPT and Math CPT settings, we train the model once for 20B tokens. We then perform individual cooldowns using intermediate checkpoints as follows: We set a target max training duration (e.g. 8 billion tokens), and define the last 20% of max training duration as the cooldown period. We then retrain from the latest available checkpoint prior to the cooldown period.

**Code CPT** Llama-2-7B trained on the StarCoder-Python dataset.

- **seq_len**: 4096
- **optimizer**: decoupled_lionw (betas=[0.9, 0.95])
- **learning_rate**: 1.0e-05 for LoRA and Full Finetuning
- **scheduler**: inv_sqrt_with_warmup (t_scale=1000ba, t_warmup=1000ba, t_cooldown=5086ba, alpha_f_decay=1, alpha_f_cooldown=0). We note that this ends up looking very much like a trapezoidal schedule.
- **weight_decay**: 1.0e-06
- **precision**: amp_bf16
- **global_train_batch_size**: 192
- **device_train_microbatch_size**: 6
- **gradient_clipping**: norm (threshold=1)
- **num_gpus**: 32
- LR Scheduler: Inverse square root with warmup $t_{warmup} = 500$ batches, $t_{scale} = 500$ batches, $t_{cooldown} = 5200$ batches $\alpha_{fdecay} = 1.0$ $\alpha_{fcooldown} = 0.0$

**Math CPT**. Llama-2-7B trained on the OpenWebMath dataset.

- **max_seq_len**: 4096
- **optimizer**: decoupled_lionw (betas=[0.9, 0.95])
- **learning_rate**: 1.0e-05 for full finetuning, 4.0e-05 for LoRA
- **scheduler**: inv_sqrt_with_warmup (t_scale=1000ba, t_warmup=1000ba, t_cooldown=5086ba, alpha_f_decay=1, alpha_f_cooldown=0). We note that this ends up looking very much like a trapezoidal schedule.
- **weight_decay**: 0
- **precision**: amp_bf16
- **global_train_batch_size**: 192
- **device_train_microbatch_size**: 6
- **gradient_clipping**: norm (threshold=1)
- **num_gpus**: 32

**Code IFT:** Finetuning Llama-2-7b on the Magicoder-Evol-Instruct-110K dataset

- **max_seq_len**: 4096
- **optimizer**: decoupled_lionw (betas=[0.9, 0.95])
- **learning_rate**: 2e-4 for rank $r = 16, 64$ and 1e-4 for $r = 256$ $\alpha = 2r = 512$ (due to instabilities/loss spikes at 2e-4)

- **scheduler**: cosine_with_warmup (alpha_f=0.01, t_warmup=0.1dur)
- **weight_decay**: 0
- **precision**: amp_bf16
- **global_train_batch_size**: 192
- **device_train_microbatch_size**: 6
- **gradient_clipping**: norm (threshold=1)
- **num_gpus**: 32

**Math IFT:** Finetuning Llama-2-7b on the MetaMathQA dataset

- **seq_len**: 1024
- **optimizer**: decoupled_lionw (betas=[0.9, 0.95])
- **learning_rate**: Full finetuning: 1e-5, LoRA: 1e-4 for $r = 16, 64$, 5e-5 for $r \approx 256$ due to instabilities.
- **scheduler**: cosine_with_warmup (alpha_f=0.01, t_warmup=0.1dur)
- **weight_decay**: 0
- **precision**: amp_bf16
- **global_train_batch_size**: 768
- **device_train_microbatch_size**: 24
- **gradient_clipping**: norm (threshold=1)
- **num_gpus**: 32

### A.1 Training the input and output embedding layers.

Vanilla LoRA and other popular methods such as QLoRA (Dettmers et al., 2024) often do not train the input and output embedding layers. Recent open-source work[11], on the other hand, shows that it might be beneficial to supplement LoRA with full finetuning of these two modules (additional $\approx 200M$ parameters for a 7B model). We view this approach as a hybrid of LoRA and full finetuning, and therefore leave its empirical investigation for future work. Moreover, this hybrid approach involves further hyperparameter optimization: the input and output layers require tuning their own separate learning rates, which should typically be 2-10X smaller than the LoRA learning rates (training with a single learning rate results in instabilities).

## B   Learning rate searches

We perform a learning rate sensitivity analysis for Llama-2-7B, trained for two epochs on the code and math IFT datasets, and followed by HumanEval and GSM8K evaluation, respectively. Fig. S1 shows that LoRA improves monotonically with learning rate up to a value at which training diverges, with best learning rates of $5e^{-4}$ for code and $2e^{-4}$ for math.

On both datasets, these best LoRA learning rates are underperformed by four alternative full finetuning learning rates. The best full finetuning learning rates are $5e^{-5}$ and $1e^{-5}$, respectively, an order of magnitude smaller than LoRA. For LoRA, we cannot find alternative learning rates that achieve at least 90% of the best learning rate's performance. For full finetuning, there are two viable alternative learning rates for code and three for math.

Note that in these experiments, the LoRA models target all modules but the $W_{\text{gate}}$, with $\alpha = 32$ which should preferably be higher for $r = 64$.

### B.1   Learning rate sensitivity analysis across optimizers

We compared the AdamW and Decoupled LionW optimizers by training for two epochs of Magicoder-Evol-Instruct-110K using different learning rates. We found that Decoupled LionW performed better on HumanEval for both LoRA and full finetuning, and across learning rates, as seen in Fig. S2.

---

[11]https://unsloth.ai/blog/contpretraining, see also the following blogpost https://www.anyscale.com/blog/fine-tuning-llms-lora-or-full-parameter-an-in-depth-analysis-with-llama-2 (Niederfahrenhorst et al., 2023)

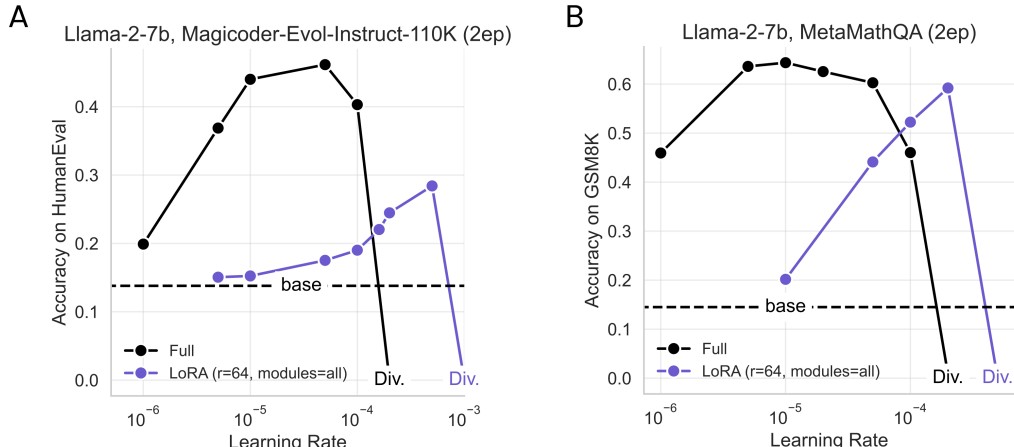

Figure S1: **LoRA is more sensitive to learning rates compared to full finetuning**. Llama-2-7B models (A) trained on *Magicoder-Evol-Instruct-110k* (Wei et al., 2023) and evaluated on HumanEval, (B) trained on *MetaMathQA* (Yu et al., 2023) and evaluated on GSM8K. Experiments here are performed with LionW; see Fig. S2 for a comparion to AdamW.

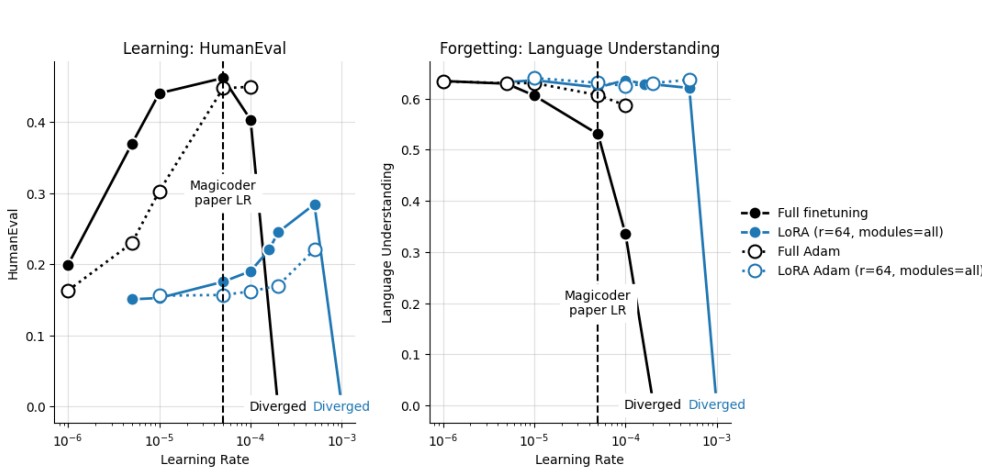

Figure S2: **Comparing LionW to AdamW across learning rates** for two epochs of the Magicoder-Evol-Instruct-110K dataset. Left: HumanEval; Right: Average of "Language Understanding" benchmarks in the MosaicML evaluation gauntlet. Both methods peak at the learning rate used in the original paper (Wei et al., 2023).

## B.2   The importance of the $\alpha$ parameter for LoRA

We found that the performance of all models was particularly sensitive to the LoRA $\alpha$ hyperparameter. Figure S3 shows two experiments on two separate datasets (Magicoder-Evol-Instruct-110K and Open Web Math) for LoRA with rank $r = 256$. In both cases the best accuracy is achieved when $\alpha = 2r$.

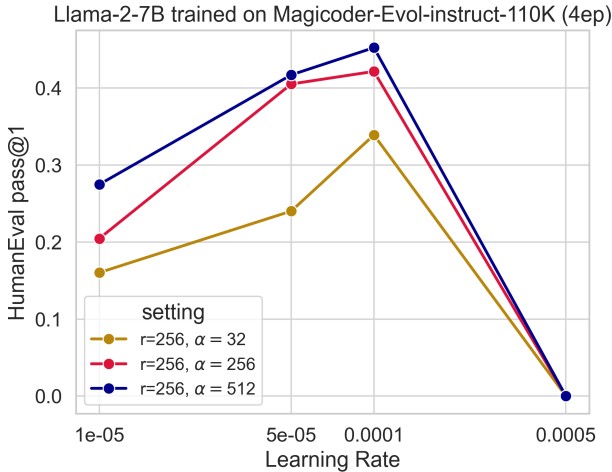
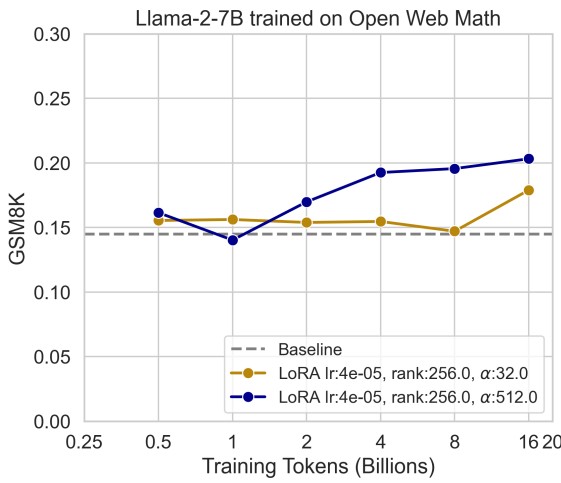

(a) Jointly sweeping over LoRA $\alpha$ and learning rate. The optimal choice is $\alpha = 2r$ (blue).

(b) Continued pretraining with two different choices of $\alpha$, where $\alpha = 2r$ is best (blue).

Figure S3: **LoRA performance is sensitive to the $\alpha$ hyperparameter**. We show that for Code IFT (a) and math CPT (b) an $\alpha$ that is scaled with rank such that $\alpha = 2r$ leads to the highest accuracy.

## C  Finetuning on the Tülu-v2-mix dataset

We finetuned Llama-2-7b models on the Tülu-v2-mix (Ivison et al., 2023), a dataset which contains a variety of finetuning datasets containing chain of thought reasoning, multi-turn assistant conversations, math and science problems, code, and more. There are roughly 326k samples in this dataset.

As in all main experiments, we compared full finetuning and LoRA $r = 16, 64, 256$, targeting all transformer modules. For each of the four experimental conditions, we trained a model for up to 6 epochs and evaluated it after 2, 4, and 6 epochs. Different from the main experiments, the checkpoints evaluated are "hot" and are not cooled down for each training duration.

As in the original paper (Ivison et al., 2023), we assess math capabilities with **GSM8K** Cobbe et al. (2021), STEM, humanities, and social science capabilities as the average of 57 subjects of **the Massive Multitask Language Understanding** (MMLU; Hendrycks et al. (2020)), and conversational capabilities with **Multi-Turn Benchmark** (MT-bench (Zheng et al., 2024)) which includes 80 multi-turn conversations where the model responses are evaluated automatically by GPT-4. We also compute the same average forgetting score as in all other datasets in this paper.

Since datasets like Tülu-v2-mix are where LoRA is mostly used, we ask: can LoRA, even with a low rank, achieve full finetuning-accuracy both in specific domains and in general conversational capabilities?

### C.1  Experimental setup

After an initial learning rate sweep, we chose the following hyperparameters:

- **max_seq_len**: 4096
- **optimizer**: decoupled_lionw (betas=[0.9, 0.95])
- **learning_rate**: Full finetuning: 5e-6; LoRA 1e-4
- **scheduler**: cosine_with_warmup (alpha_f=0.01, t_warmup=0.1dur)
- **weight_decay**: 0
- **precision**: amp_bf16
- **global_train_batch_size**: 192
- **device_train_microbatch_size**: 6
- **gradient_clipping**: norm (threshold=1)
- **num_gpus**: 32

### C.2  Results

First, we find that on MT-bench (Fig. S5), both LoRA and full finetuning meaningfully improve upon the base model (2.74), starting from the second epoch and improving only slightly when trained for longer. Crucially, all LoRA models are within one standard error of the mean of the full finetuning model (computed with 160 datapoints = 80 questions × 2 turns). That is, one can achieve full finetuning conversational capabilities with $r = 16$. The caveat is that only 80 questions appear in this benchmark and that the variance, within model, is high.

In GSM8K (Fig. S6a), again, all models significantly improve upon the base model (0.145). Remarkably, even in this specific domain, LoRA and full finetuning are overlapping, with the best model being LoRA $r = 256$ at epoch 4, which is followed by full finetuning at epoch 2. Here too, as in the other math datasets in the paper, there is an ordering by LoRA rank.

In MMLU (Fig. S6b), full finetuning and LoRA are overlapping with LoRA $r = 64$ as the best model (epoch 4), followed by full finetuning at epoch 2. Here there is no ordering by rank.

As for forgetting (Fig. S7), we find an overall mild forgetting compared to the rest of the datasets in the paper. At two epochs, full finetuning does better than LoRA. The former starts to degrade at epoch 4. At epoch 6, the findings of the main paper are replicated: full finetuning forgets the most and we find a clear ordering of forgetting by rank.

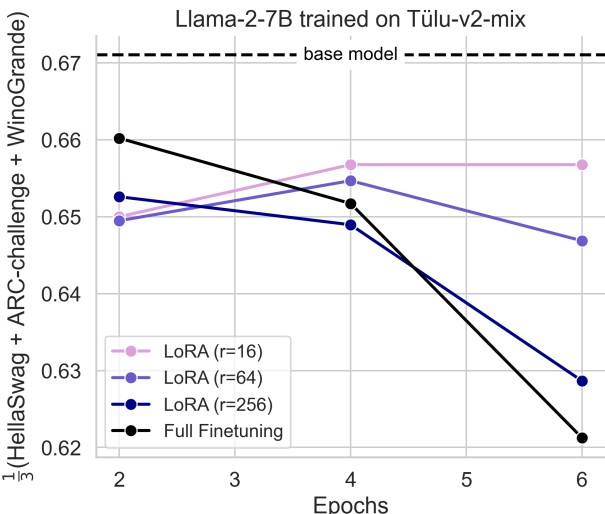

Figure S4: **LoRA forgets less even on a more diverse IFT dataset like Tülu-v2-mix.** We plot the average forgetting score, same as in all other datasets, as a function of training duration.

Across all evaluations – learning and forgetting – full finetuning is the best model at epoch 2, and only degrades afterwards. LoRA, on the other hand, needs 4 epochs to train, mirroring the findings in the main part of the paper. LoRA $r = 16$ seems to offer competitive conversational capabilities, and minimal forgetting, but it underperforms in domain-specific knowledge like math. Future work should seek to understand why this is the case.

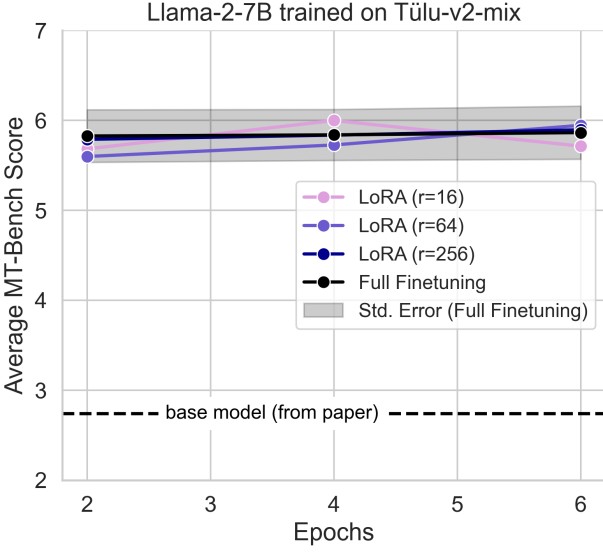

Figure S5: **Average MT-bench score with GPT-4 as a judge, calculated over 80 questions with two turns each**. Base model value as reported in the MT-bench paper. We note that the Tülu paper reports a 6.3 MT-bench value from full finetuning of Llama-2-7b base model, which is only slightly exceeding the standard error from our average score.

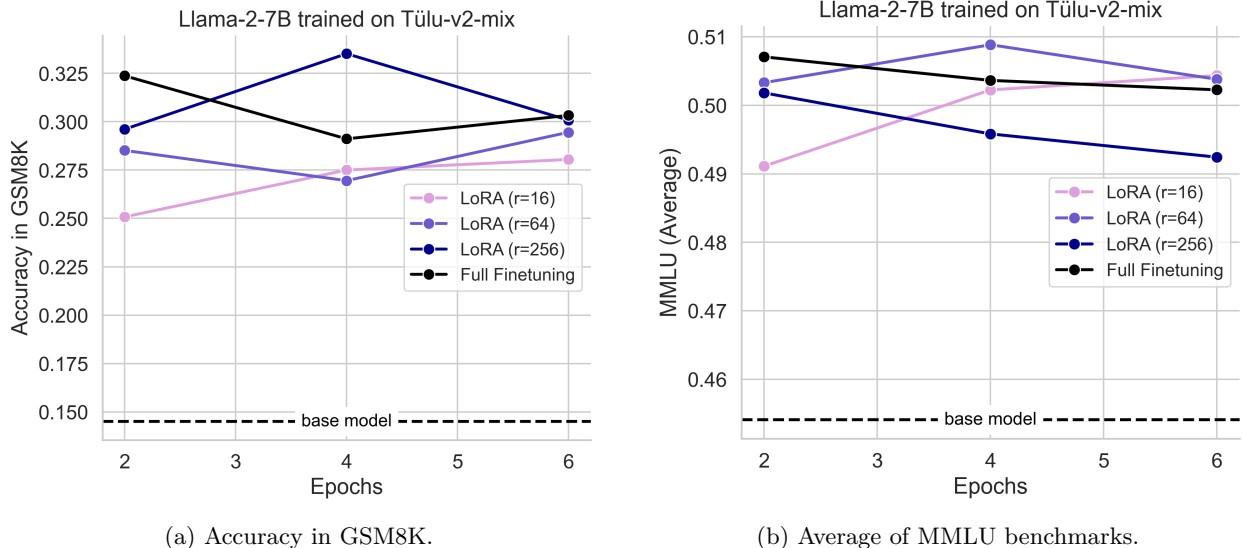

(a) Accuracy in GSM8K.

(b) Average of MMLU benchmarks.

Figure S6: **On Tülu-v2-mix, LoRA and full finetuning both improve upon the base model and perform comparably.**

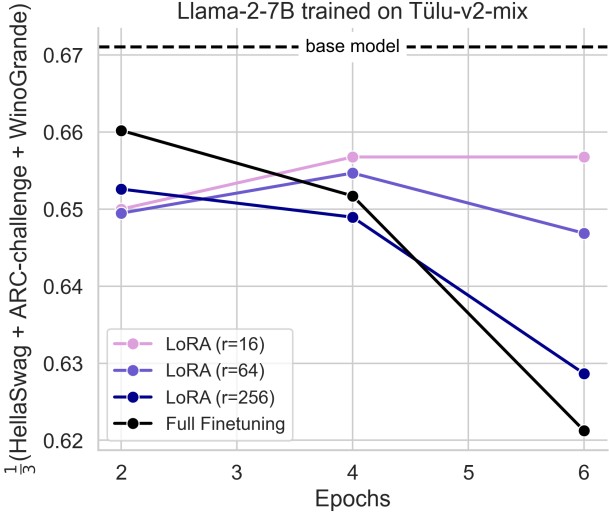

Figure S7: **LoRA forgets less even on a more diverse IFT dataset like Tülu-v2-mix.** We plot the average forgetting score, same as in all other datasets, as a function of training duration.

# D  Supplementary tables

Table S1: Starcoder-Python Results (HumanEval pass@1)

| Num. tokens (billions) Condition | 0.25 | 0.50 | 1 | 2 | 4 | 8 | 16 | 20 |
|---|---|---|---|---|---|---|---|---|
| LoRA (r=16) | 0.143 | 0.144 | 0.141 | 0.141 | 0.154 | 0.159 | 0.162 | 0.162 |
| LoRA (r=64) | 0.142 | 0.146 | 0.141 | 0.153 | 0.157 | 0.176 | 0.194 | 0.196 |
| LoRA (r=256) | 0.144 | 0.142 | 0.143 | 0.159 | 0.159 | 0.208 | 0.211 | 0.224 |
| Full Finetuning | 0.152 | 0.153 | 0.172 | 0.181 | 0.218 | 0.258 | 0.255 | 0.263 |

Table S2: Starcoder-Python Results (Forgetting Average)

| Num. tokens (billions) Condition | 0.25 | 0.50 | 1 | 2 | 4 | 8 | 16 | 20 |
|---|---|---|---|---|---|---|---|---|
| LoRA (r=16) | 0.645 | 0.642 | 0.645 | 0.642 | 0.644 | 0.640 | 0.638 | 0.635 |
| LoRA (r=64) | 0.646 | 0.644 | 0.646 | 0.646 | 0.639 | 0.634 | 0.626 | 0.626 |
| LoRA (r=256) | 0.644 | 0.645 | 0.643 | 0.639 | 0.636 | 0.630 | 0.618 | 0.617 |
| Full Finetuning | 0.625 | 0.624 | 0.625 | 0.616 | 0.599 | 0.583 | 0.551 | 0.545 |

Table S3: OpenWebMath Results (GSM8K)

| Num. tokens (billions) Condition | 0.25 | 0.50 | 1 | 2 | 4 | 8 | 16 | 20 |
|---|---|---|---|---|---|---|---|---|
| LoRA (r=16) | 0.162 | 0.157 | 0.161 | 0.155 | 0.165 | 0.156 | 0.152 | 0.158 |
| LoRA (r=64) | 0.163 | 0.167 | 0.150 | 0.166 | 0.164 | 0.168 | 0.179 | 0.163 |
| LoRA (r=256) | 0.162 | 0.161 | 0.140 | 0.170 | 0.193 | 0.196 | 0.203 | 0.202 |
| Full Finetuning | 0.155 | 0.152 | 0.165 | 0.158 | 0.224 | 0.238 | 0.283 | 0.293 |

Table S4: OpenWebMath Results (Forgetting Average)

| Num. tokens (billions) Condition | 0.25 | 0.50 | 1 | 2 | 4 | 8 | 16 | 20 |
|---|---|---|---|---|---|---|---|---|
| LoRA (r=16) | 0.640 | 0.641 | 0.646 | 0.641 | 0.643 | 0.641 | 0.636 | 0.637 |
| LoRA (r=64) | 0.640 | 0.640 | 0.638 | 0.637 | 0.643 | 0.634 | 0.634 | 0.627 |
| LoRA (r=256) | 0.638 | 0.638 | 0.637 | 0.634 | 0.633 | 0.620 | 0.620 | 0.616 |
| Full Finetuning | 0.634 | 0.634 | 0.640 | 0.630 | 0.629 | 0.619 | 0.613 | 0.618 |

Table S5: Magicoder-Evol-Instruct-110K Results (HumanEval pass@1)

| Epoch Condition | 1 | 2 | 4 | 8 | 16 |
|---|---|---|---|---|---|
| LoRA (r=16) | 0.197 | 0.275 | 0.358 | 0.338 | 0.324 |
| LoRA (r=64) | 0.249 | 0.339 | 0.417 | 0.392 | 0.405 |
| LoRA (r=256) | 0.299 | 0.385 | 0.498 | 0.437 | 0.466 |
| Full Finetuning | 0.302 | 0.464 | 0.470 | 0.497 | 0.416 |

Table S6: Magicoder-Evol-Instruct-110K Results (Forgetting Average)

| Epoch Condition | 1 | 2 | 4 | 8 | 16 |
|---|---|---|---|---|---|
| LoRA (r=16) | 0.653 | 0.648 | 0.652 | 0.646 | 0.609 |
| LoRA (r=64) | 0.652 | 0.651 | 0.632 | 0.580 | 0.510 |
| LoRA (r=256) | 0.655 | 0.659 | 0.631 | 0.552 | 0.517 |
| Full Finetuning | 0.595 | 0.579 | 0.512 | 0.446 | 0.414 |

Table S7: MetaMathQA Results (GSM8K)

| Epoch Condition | 1 | 2 | 4 | 8 | 16 |
|---|---|---|---|---|---|
| LoRA (r=16) | 0.447 | 0.528 | 0.580 | 0.578 | 0.569 |
| LoRA (r=64) | 0.527 | 0.588 | 0.624 | 0.624 | 0.595 |
| LoRA (r=256) | 0.557 | 0.607 | 0.625 | 0.634 | 0.594 |
| Full Finetuning | 0.604 | 0.641 | 0.642 | 0.619 | 0.599 |

Table S8: MetaMathQA Results (Forgetting Average)

| Epoch Condition | 1 | 2 | 4 | 8 | 16 |
|---|---|---|---|---|---|
| LoRA (r=16) | 0.628 | 0.617 | 0.616 | 0.616 | 0.596 |
| LoRA (r=64) | 0.617 | 0.609 | 0.608 | 0.586 | 0.568 |
| LoRA (r=256) | 0.613 | 0.607 | 0.599 | 0.584 | 0.567 |
| Full Finetuning | 0.598 | 0.599 | 0.590 | 0.572 | 0.559 |

Table S9: Tülu-v2-mix Results

Table S10: Tülu-v2-mix MT-Bench

| Epoch
Condition | 2 | 4 | 6 |
|---|---|---|---|
| LoRA (r=16) | 5.681 | 5.997 | 5.712 |
| LoRA (r=64) | 5.597 | 5.725 | 5.944 |
| LoRA (r=256) | 5.788 | 5.834 | 5.894 |
| Full Finetuning | 5.825 | 5.838 | 5.862 |

Table S11: Tülu-v2-mix MMLU

| Epoch
Condition | 2 | 4 | 6 |
|---|---|---|---|
| LoRA (r=16) | 0.491 | 0.502 | 0.504 |
| LoRA (r=64) | 0.503 | 0.509 | 0.504 |
| LoRA (r=256) | 0.502 | 0.496 | 0.492 |
| Full Finetuning | 0.507 | 0.504 | 0.502 |

Table S12: Tülu-v2-mix GSM8K

| Epoch
Condition | 2 | 4 | 6 |
|---|---|---|---|
| LoRA (r=16) | 0.251 | 0.275 | 0.280 |
| LoRA (r=64) | 0.285 | 0.270 | 0.295 |
| LoRA (r=256) | 0.296 | 0.335 | 0.301 |
| Full Finetuning | 0.324 | 0.291 | 0.303 |

Table S13: Tülu-v2-mix Forgetting Average

| epoch
condition | 2 | 4 | 6 |
|---|---|---|---|
| LoRA (r=16) | 0.650 | 0.657 | 0.657 |
| LoRA (r=64) | 0.649 | 0.655 | 0.647 |
| LoRA (r=256) | 0.653 | 0.649 | 0.629 |
| Full Finetuning | 0.660 | 0.652 | 0.621 |

# E    Supplementary Figures for SVD Analysis

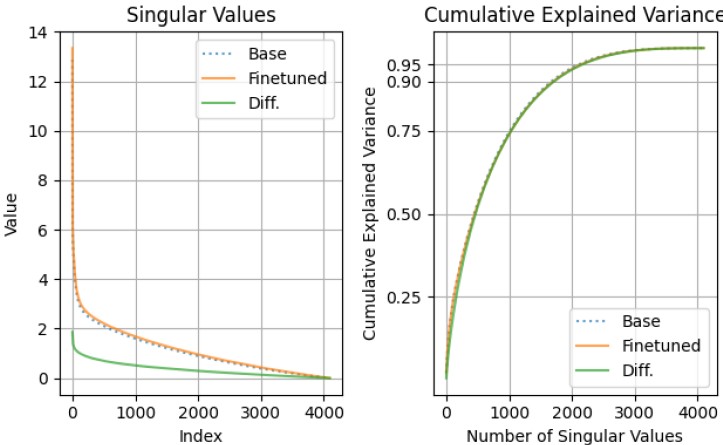

Figure S8: **SVD analysis for** $4096 \times 4096$ **matrix** $W_q$ **at layer 26**. Left: singular values for base weights, finetuned weights, and their difference. Right: cumulative explained variance. Notice that for all three matrices, a rank $> 1500$ is needed to explain 90% of the variance.

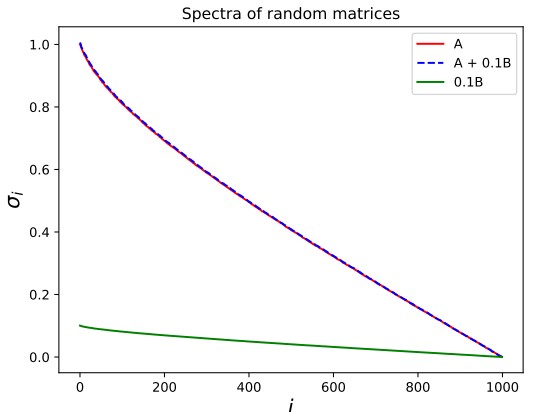

(a) Spectrum of $A$ and $A + cB$ as well as $cB$ for $c = 0.1$. Notably, $A, cB, A + cB$ are all high rank.

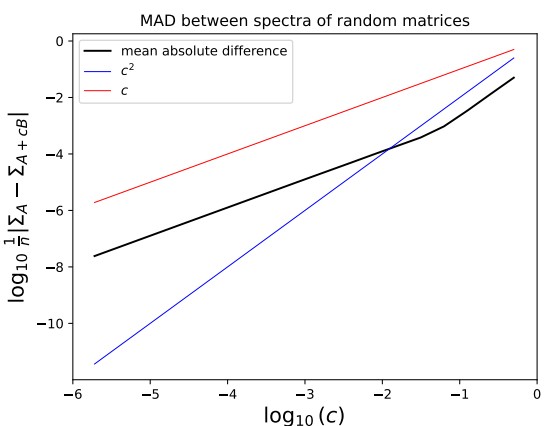

(b) Mean absolute difference between spectra of $A$ and $A + cB$ for various $c$.

Figure S9: **Analyzing the spectra of the sum of two** $1000 \times 1000$ **Gaussian i.i.d matrices**. $A$ and $B$ are $1000 \times 1000$ random matrices with i.i.d. standard normal Gaussian entries.

# F  Solution Generation Diversity on HumanEval

For the best set of Llama-2-7B models trained in MagicCoder we evaluate how their pass@$k$ metric in the HumanEval benchmark increases as we increase the parameter $k$ which controls the acceptance criterion. The pass@$k$ metric (Chen et al., 2021) is defined as

$$\text{pass@}k := \mathbb{E}\left[1 - \frac{\binom{n-c}{k}}{\binom{n}{k}}\right],\qquad(1)$$

where $n$ is the number of generations, $c$ the number of correct generations and $k$ determines the size of the sample set of generations considered for acceptance. Assuming we sample from the model outputs, i.e. sampling temperature $T > 0$, then increasing $k$ will increase the diversity of generations, and increase the likelihood of a passing generation being present in a random subset of size $k$.

Figure S10 reports pass@$k$ for the LoRA models trained in the MagicCoder dataset as well as the base Llama-2-7B model. For all models, as we increase $k$, the pass@$k$ consistently and monotonically improves. Finetuned models scores are substantially higher than the base model. At $k = 1$, Full finetuning outperforms the LoRA model whose scores are ordered from largest to smallest rank, as expected. As $k$ increases we observe all models improving their pass@$k$ scores and the gap between them reducing when $k > 16$. We note that full finetuning is superior across all values of $k$ with temperature 0.8. This complements the results in 1 which used a temperature of 0.2 and pass@1, where the improvements upon $r = 256$ at epoch 4 are less clear.

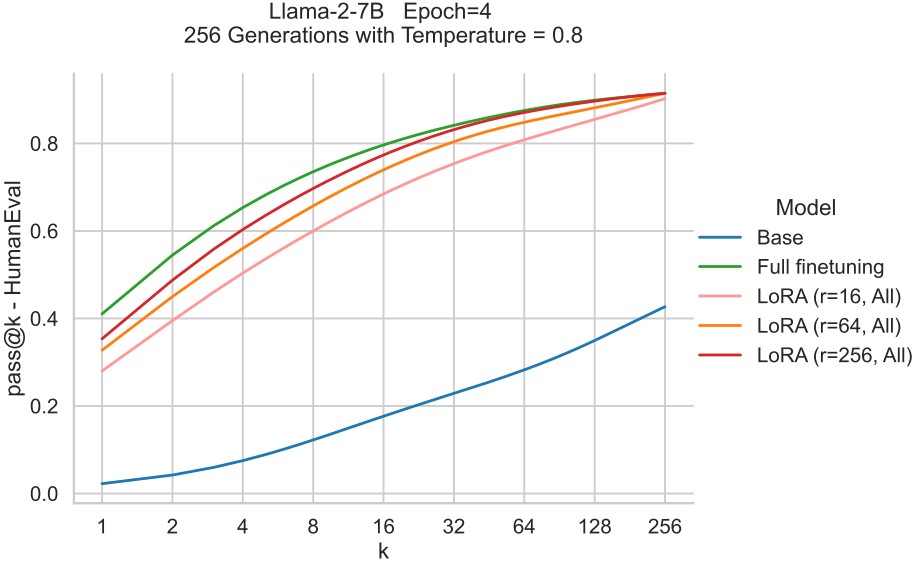

Figure S10: **HumanEval pass@$k$ for models trained on the Magicoder dataset.** For every model, we sample 256 independent generations with temperature 0.8.

# G Training Datasets

## G.1 MetaMathQA (Math IFT)

The MetaMathQA dataset (Yu et al. (2023), `https://huggingface.co/datasets/meta-math/MetaMathQA`) contains 395,000 samples that are bootsrapped from the GSM (Cobbe et al., 2021) and Math (Hendrycks et al., 2021) training sets. These samples are augmented by GPT-3.5 using the following methods:

- Answer Augmentation (155k samples, Yu et al. (2023)): this method proposed by the MetaMathQA authors generates multiple reasoning paths for a given mathetical question and filters for generated reasoning paths that contain the correct final answer.

- Rephrasing (130k samples, (Yu et al., 2023)): this method proposed by the MetaMathQA authors uses GPT-3.5 to rephrase questions. They check for the correctness of rephrased questions by using few-shot Chain of Thought prompting to compare reasoning chains and proposed answers with ground truth answers.

Both Self-Verification (Weng et al., 2022) and FOBAR (Jiang et al., 2024b) fall under the category of "backward reasoning," where the question starts with a given condition and requires reasoning backwards to solve for an unknown variable. In order to generate new mathematical questions, a numerical value in the original question is masked as a variable X, and the question is rephrased accordingly.

- Self-Verification (55k samples, Weng et al. (2022)): the question is rephrased into a declarative statement followed by a question about the masked variable X.

- FOBAR (55k samples, Jiang et al. (2024b)): this approach is similar to Self-Verification but directly appends the answer to the question using the template "If we know the answer to the above question is $\hat{A}_c$, what is the value of unknown variable $x$?"

MetaMathQA samples are organized by 4 columns: `type`, `original_question`, `query` and `response`.

We include two full examples below:

> **Example G.1: MetaMathQA**
>
> `type:` GSM_Rephrased
>
> `query:` Scott wants to prepare and freeze a large portion of ratatouille. He purchases 5 pounds of eggplants and 4 pounds of zucchini at a rate of $2.00 per pound from the farmers' market. Additionally, he needs 4 pounds of tomatoes that cost $3.50 per pound, 3 pounds of onions priced at $1.00 per pound, and a pound of basil sold for $2.50 per half pound. If this quantity yields 4 quarts, what is the cost per quart?
>
> `original_question:` Scott wants to make and freeze a large batch of ratatouille. At the farmers' market he buys 5 pounds of eggplants and 4 pounds of zucchini at $2.00 a pound. He needs 4 pounds of tomatoes that are $3.50 a pound. The onions are $1.00 a pound and he needs 3 pounds. Then he needs a pound of basil which is sold for $2.50 per half pound. If this yields 4 quarts, how much does each quart cost?
>
> `response:` The cost of the eggplants is 5 pounds x $2.00 per pound = $10.00. The cost of the zucchini is 4 pounds x $2.00 per pound = $8.00. The cost of the tomatoes is 4 pounds x $3.50 per pound = $14.00. The cost of the onions is 3 pounds x $1.00 per pound = $3.00. The cost of the basil is 1 pound x ($2.50 per half pound x 2) = $5.00. The total cost for all the ingredients is $10.00 + $8.00 + $14.00 + $3.00 + $5.00 = $40.00. Since this quantity yields 4 quarts, the cost per quart is $40.00 / 4 quarts = $10.00 per quart. #### 10 The answer is: 10

---

**Example G.2: MetaMathQA**

`type`: MATH_SV

`query`: If a school bus leaves school with X students on board, and one-half of the students get off the bus at each of the first three stops, The number of students remaining on the bus after the third stop is 6. What is the value of unknown variable X?

`original_question`: If a school bus leaves school with 48 students on board, and one-half of the students get off the bus at each of the first three stops, how many students remain on the bus after the third stop?

`response`: To solve this problem, we need to determine the value of x, which represents the initial number of students on the bus. Let's break down the information given: Number of students remaining on the bus after the third stop: 6 Number of students getting off the bus at each stop: 1/2 Number of stops: 3 We can set up the equation as follows: Number of students on the bus * $(1/2)^3$ = Number of students remaining on the bus after the third stop x * $(1/2)^3$ = 6 Let's simplify and solve for x: x * (1/8) = 6 To isolate x, we multiply both sides of the equation by 8: x * (1/8) * 8 = 6 * 8 x = 48 The value of x is 48. The answer is: 48

---

## G.2 Magicoder-Evol-Instruct-110k (Code IFT)

---

**Example G.3: Magicoder-Evol-Instruct-110k**

`instruction`: Please amend the subsequent Python script so that it includes a 'while' loop rather than the existing 'for' loop, which iterates through the items of an integer list. The script currently has a bug where it attempts to print an object that is outside the bounds of the list. Fix this error and modify the script to use 'while' instead of 'for' loop. Ensure your script correctly handles empty lists.

```python
# Establish an integer list
arr = [1, 2, 3, 4]
# Determine the length of the list
n = len(arr)
# Traverse the list and output each individual element
for i in range(n+1): print(arr[i]) ```
```

`response`:

```python
# Establish an integer list arr = [1, 2, 3, 4]
# Determine the length of the list
n = len(arr)
# Initialize index at 0 i = 0
# Traverse the list and output each individual element
while i < n: print(arr[i]) i += 1
```

In the given code, it tries to access 'arr[n]' which is out of bounds as python list indexing starts from 0 and ends at n-1 for a list of length n. Also, I used 'while' loop as per your request. By initializing 'i=0' before the loop and 'i += 1' within the loop, we can iterate over the list. This iteration continues until 'i' is less than 'n', the length of the list. The 'while' loop will not proceed if the list is empty so it correctly handles empty lists.

### G.3   Starcoder Python (Code CPT)

**Example G.4: Starcoder-Python**

```python
"""
function of invoking Gitee API
"""

import base64
import logging
import requests
from flask import current_app
from requests import exceptions
logger = logging.getLogger(__name__)

ORG_URL = "\href{https://gitee.com/api/v5/orgs}{https://gitee.com/api/v5/orgs}"

REPO_URL = "\href{https://gitee.com/api/v5/repos}{https://gitee.com/api/v5/repos}"

def get_request(url, params):

    """

    get request

    """

    logger.debug("Get request, connect url: %s", url)

    try:

        response = requests.get(url,params=params)

        return True, response

    except exceptions.ConnectionError as err:

        logger.error(err)

        return False, 'connection error'

    except IOError as err:

        logger.error(err)

        return False, 'IO error'
```

more functions truncated...

### G.4 OpenWebMath (Math CPT)

---

**Example G.5: OpenWebMath**

`url:` `http://math.stackexchange.com/questions/222974/probability-of-getting -2-aces-2-kings-and-1-queen-in-a-five-card-poker-hand-pa`

`text:` # Probability of getting 2 Aces, 2 Kings and 1 Queen in a five card poker hand (Part II) So I reworked my formula in method 1 after getting help with my original question - Probability of getting 2 Aces, 2 Kings and 1 Queen in a five card poker hand. But I am still getting results that differ...although they are much much closer than before, but I must still be making a mistake somewhere in method 1. Anyone know what it is? Method 1 $P(2A \cap 2K \cap 1Q) = P(Q|2A \cap 2K)P(2A|2K)P(2K)$ $$= \frac{1}{12}\frac{{4 \choose 2}{46 \choose 1}}{50 \choose 3}\frac{{4 \choose 2}{48 \choose 3}}{52 \choose 5}$$ $$= \frac{(6)(17296)(6)(46)}{(2598960)(19600)(12)}$$ $$= 4.685642 * 10\hat{-5}$$ Method 2 $$\frac{{4 \choose 2} {4 \choose 2}{4 \choose 1}}{52 \choose 5} = \frac{3}{54145}$$ $$5.540678 * 10\hat{-5}$$ - Please make an effort to make the question self-contained and provide a link to your earlier question.
– Sasha Oct 28 '12 at 19:56 I think we would rather ahve you edit your initial question by adding your new progress. This avoids having loss of answer and keeps track of progress – Jean-Sébastien Oct 28 '12 at 19:56 But there already answers to my original question so those answers would not make sense now that I am using a new formula for method 1.
– sonicboom Oct 28 '12 at 20:03 Conditional probability arguments can be delicate. Given that there are exactly two Kings, what's the $46$ doing? That allows the possibility of more Kings.
– André Nicolas Oct 28 '12 at 20:26 The $46$ is because have already taken two kings from the pack leaving us with 50. And now we have chosen 2 aces and we have to pick the other 1 card from the 50 remaining cards less the 4 aces?
– sonicboom Oct 28 '12 at 20:42 show 1 more comment $$\frac{1}{11}\frac{{4 \choose 2}{44 \choose 1}}{48 \choose 3}\frac{{4 \choose 2}{48 \choose 3}}{52 \choose 5}$$ If you wrote this as $$\frac{{4 \choose 2}{48 \choose 3}}{52 \choose 5}\frac{{4 \choose 2}{44 \choose 1}}{48 \choose 3}\frac{{4 \choose 1}{40 \choose 0}}{44 \choose 1}$$ it might be more obvious why they are the same.

`date:` `2014-03-07 11:01:44`

---

There is often some confusion about the memory gains that vanilla LoRA offers both in theory and in practice. In Appendix H we discuss some of the theoretical benefits of LoRA, and show how it can enable training both on GPUs with less memory and on fewer total GPUs (in the multi-GPU setting). In Appendix I we show how LoRA in practice leads to memory savings relative to full finetuning, but can in fact lead to slower throughput for particular hardware and software settings.

## H   Theoretical Memory Efficiency Gains with LoRA for Single and Multi-GPU Settings

Modern systems for training neural networks store and operate on the following objects (following the conventions in Rajbhandari et al. (2020)). Most memory requirements relate to *model states*, which include:

- parameter weights
- gradients
- higher order optimization quantities such as optimizer momentum and variance in the Adam optimizer, and the momentum in the Lion optimizer

The remaining memory requirements come from the *residual states*:

- activations (which depend on batch size and maximum sample sequence length)
- temporary buffers for intermediate quantities in the forward and backward pass.

which will require more memory when increasing the batch size and maximum sequence lengths.

LoRA offers memory savings with respect to the *model states*. The next two sections describe these memory savings in the single GPU and multi-GPU setting with examples loosely inspired by Rajbhandari et al. (2020).

The data stored at single precision includes:

- a "master copy" of the tuned parameter weights
- the gradient
- all optimizer states (both momentum and variance for Adam, and just momentum for Lion)

For simplicity, we do not consider mixed-precision training, which involves storing critical data at single precision (fp32; 4 bytes per number) while performing some computations at half precision (fp16 or bfloat16; 2 bytes per number).

### H.1   Training on a Single GPU

In the single GPU setup, the difference in memory requirements between LoRA and full finetuning is particularly drastic when using the Adam optimizer (Hu et al., 2021; Rajbhandari et al., 2020).

Storing the master weights in fp32 requires 4 bytes per parameter, while storing the gradient in fp32 requires 4 bytes *per tuned parameter*. In order to maintain the optimizer state in fp32 for Adam, 8 bytes per tuned parameter are required; 4 bytes for the momentum term, and 4 bytes for the variance term. Let $\Psi$ be the number of model parameters. Therefore, in the Adam full finetuning setting of a $\Psi = 7B$ parameter model, the total memory requirements are at least roughly $4 \times \Psi + 4 \times \Psi + 8 \times \Psi = 112$ GB.

The Lion optimizer only uses a momentum term in the gradient calculation, and the variance term in Adam therefore disappears. In the Lion full finetuning setting of a $\Psi = 7B$ parameter model, the total memory requirements are therefore roughly $4 \times \Psi + 4 \times \Psi + 4 \times \Psi = 84$ GB.

LoRA, on the other hand, does not calculate the gradients or maintain optimizer states (momentum and variance terms) *for most of the parameters*. Therefore the amount of memory used for these terms is drastically reduced.

| 7B Training | 1 GPU | 8 GPUs | 16 GPUs | 32 GPUs | 64 GPUs |
|---|---|---|---|---|---|
| Adam | 112 GB | 14 GB | 7 GB | 3.5 GB | 1.75 GB |
| Adam + LoRA | 15.12 GB | 1.89 GB | 0.945 GB | 0.4725 GB | 0.236 GB |
| Lion | 84 GB | 10.5 GB | 5.25 GB | 2.625 GB | 1.3125 GB |
| Lion + LoRA | 14.84 GB | 1.855 GB | 0.9275 GB | 0.464 GB | 0.232 GB |

Table S14: **Theoretical memory required to store the model and optimizer state during training for a 7B parameter model.** Note that the numbers exclude memory needed to store activations. FSDP sharding the parameter and optimizer states across N devices results in less memory usage relative to LoRA. LoRA on the other hand enables training on GPUs with far less memory and also enables training without needing as many GPUs to shard across.

A LoRA setting with Adam that only tunes matrices that are 1% of the total parameter count (e.g. $\Psi = 7B$ base model with 70M additional parameters used by LoRA) requires roughly $4 \times \Psi(1 + 0.01) + 4 \times \Psi \times 0.01 + 8 \times \Psi \times 0.01 = 29.12$ GB of memory. Theoretically this can be reduced further to $2 \times \Psi + 16 \times \Psi \times 0.01 = 15.12$ GB *if the non-tuned parameter weights are stored in bfloat16*. We use this assumption for the subsequent examples.

Note again that these numbers do not take into consideration sample batch size or sequence length, which affect the memory requirements of the activations.

### H.2 Training on a Multiple with Fully Sharded Data Parallelism

Past approaches for training LLMs across multiple GPUs include model parallelism, where different layers of the LLM are stored on different GPUs. However this requires high communication overhead and has very poor throughput (Rajbhandari et al., 2020). Fully Sharded Data Parallelism (FSDP) shards the parameters, the gradient, and the optimizer states across GPUs. This incredibly efficient and actually is competitive with the memory savings offered by LoRA in certain settings.

FSDP sharding the parameter and optimizer states across N devices results in less memory usage relative to LoRA. LoRA on the other hand enables training on GPUs with far less memory and also emanes training without needing as many GPUs to shard across.

For example, in the Adam full finetuning setting of a $\Psi = 7B$ parameter model on 8 GPUs with FSDP, the total memory requirement for *each* GPU is roughly $(4 \times \Psi + 4 \times \Psi + 8 \times \Psi)/8 = 14$ GB. This reduces further to 3.5 GB for FSDP with 32 GPUs (see Table S14).

The LoRA with Adam setup on 8 GPUs (where $\Psi = 7B$ base model and there are 70M additional LoRA parameters) requires roughly $(2 \times \Psi + 16 \times \Psi \times 0.01)/8 = 1.89$ GB of memory per GPU. With 32 GPUs this decreases further to 0.4725 GB.

Standard industry level GPUs have on-device memory between 16 GB (e.g. V100s) and 80 GB (e.g. A100s and H100s). As Table S14 demonstrates, the per-GPU memory requirements for training a 7B parameter model decrease drastically as the number of GPUs increase. The memory requirements for training a 7B model with Adam + LoRA on a single GPU are 15.12 GB, but the same per-GPU memory requirement for training a 7B model with Adam but *without* LoRA on 8 GPUs is 14 GB. In this 8 GPU scenario, the efficiency gains from LoRA disappear.

Table S15 applies similar calculations to a 70B parameter model. Finetuning such a large model on 8 GPUs is *only* possible using a technique like LoRA; where Adam requires 140 GB per GPU, Adam+LoRA requires 18.9 GB per GPU. The efficiency gains of LoRA relative to FSDP therefore depend on the model size and GPU availability/cost considerations.

We do the same analysis for a 405B parameter model to highlight how LoRA is beneficial as model size scales (Table S16).

| 70B Training | 1 GPU | 8 GPUs | 16 GPUs | 32 GPUs | 64 GPUs |
|---|---|---|---|---|---|
| Adam | 1.12 TB | 140 GB | 70 GB | 35 GB | 17.5 GB |
| Adam + LoRA | 151.2 GB | 18.9 GB | 9.45 GB | 4.725 GB | 2.36 GB |
| Lion | 840 GB | 105 GB | 52.5 GB | 26.25 GB | 13.125 GB |
| Lion + LoRA | 148.4 GB | 18.55 GB | 9.275 GB | 4.64 GB | 2.32 GB |

Table S15: **Theoretical memory required to store the model and optimizer state during training for a 70B parameter model.**

| 405B Training | 1 | 8 | 16 | 32 | 64 | 128 | 256 |
|---|---|---|---|---|---|---|---|
| Adam | 6480 | 810 | 405 | 202.5 | 101.25 | 50.625 | 25.3 |
| Adam + LoRA | 874.8 | 109.35 | 54.65 | 27.34 | 13.67 | 6.83 | 3.42 |
| Lion | 4860 | 607.5 | 303.75 | 151.875 | 75.94 | 37.97 | 18.98 |
| Lion + LoRA | 858.6 | 107.325 | 53.66 | 26.83 | 13.42 | 6.71 | 3.35 |

Table S16: **Theoretical memory required to store the model and optimizer state during training for a 405B parameter model.** Units are in gigabytes (GB)

## I  LoRA Throughput and Memory Measurements

We report training efficiency comparisons between full finetuning and models trained with LoRA for various of choices of rank. We measured both the throughput (in tokens per second) and peak active memory (in GB) for training runs representative of the experiments reported in the paper. We performed the runs using single node of 8×H100-80GB GPUs. We used a per-GPU micro batch size of 1 and targeted all linear layer weights with LoRA (i.e. both Attention and MLP).

In Figure S11 we observe that there is a significant gap between full finetuning and LoRA runs, related to the additional overheads of the LoRA computations. In general, **LoRA leads to an approximately** 15% **reduction in throughput** for a given batch size. LoRA with higher ranks is slower than lower ranks across all batch sizes; this is particularly noticeable for rank $r = 512$. Similarly, LoRA settings with higher batch sizes have slightly higher throughput relative to lower batch sizes. Some of the slowdown is intrinsically related to the overheads of performing LoRA, since in practice it involves more computations of intermediate activations. However, we note that we did not optimize the LoRA implementation and used the publicly available HuggingFace `peft` library, which might be amenable to further optimizations that could reduce the gap in throughput.

For peak memory, we notice that **for small batch sizes, LoRA provides a substantial reduction in peak memory** ($\sim 40\%$). This is expected since the optimizer state is significantly smaller when using parameter efficient methods. However, as batch size increases, the size of intermediate activations increases proportionally, dominating the required memory. We limit the per GPU micro batch size to 8 to prevent out of memory errors, so for batch sizes 64 and above, we perform gradient accumulation. This leads to the throughput and memory stabilizing for batch size 64 and above, with just around ($\sim 15\%$ memory savings) for larger batch sizes.

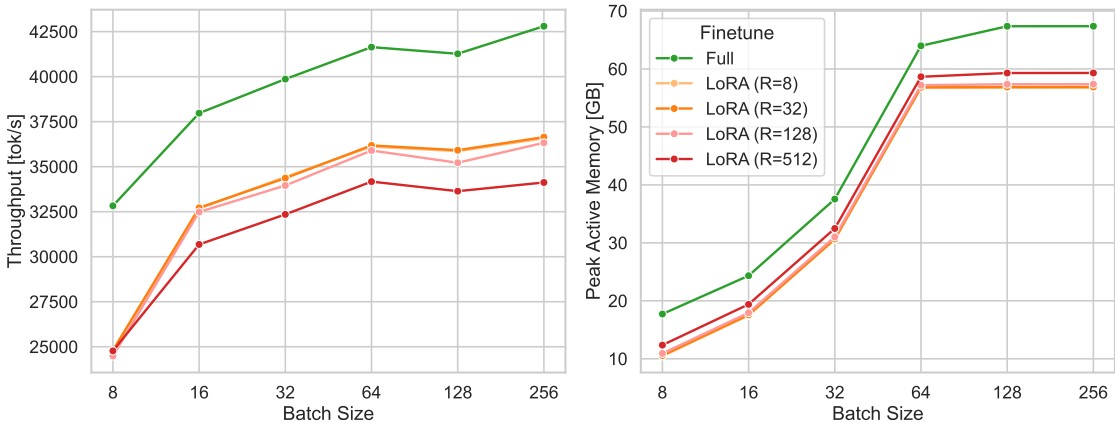

Figure S11: **Throughput and Memory Measurements for LoRA vs. full finetuning**. (left) Training throughput measured in tokens per second across all 8 GPUs. (right) Peak active memory used by the training process in a single GPU (max GPU memory is 80GB).

