# OpenReview forum: "LoRA Learns Less and Forgets Less"
_TMLR — Accepted by TMLR_

### Review · Reviewer_eciD · 2024-05-31

**Summary Of Contributions:**

This work empirically investigate the effectiveness of Low-rank based finetuning of language models for code and maths domains. Authors look at performance on old domains under domain shift (forgetting) versus task performance+generation diversity (learning). The baseline each time is full finetuning of the LLM. Their investigation focuses on Llama 2 models of different sizes only.

The authors find that LoRA underperforms full finetuning in code generation, but that it does mitigate forgetting, confirming the title of the paper. In some rare cases, LoRA-trained models sometimes perform just as well as full-finetuned models in the target domain, while forgetting less about the source domain. However, these outliers are not sufficient to say that LoRA learns just as much while forgetting less.

The work also compares the regularization effect of LoRA to regularization techniques such as dropout, and find that LoRA compares favorably, also resulting in more diverse outputs than full finetuning.

The authors hypothesize that the low-rank restriction proved effective in other tasks because the tasks were simple, and show hints that this holds by comparing the ranks of weights of LoRA-finetuned models to those of full finetuning models. Finally, they make concrete recommendations for finetuning with LoRA, namely to use it for instruction fine-tuning but to favor full finetuning for continued pretraining.

**Audience:**

Yes

**Broader Impact Concerns:**

I dont believe there are ethical implications to this work that would require a more detailed impact statement.

**Claims And Evidence:**

Yes

**Requested Changes:**

- Please provide more information about the experimental setup, especially the motivation for using different LoRA configurations between maths and code settings. Are hyperparameter choices motivated by the original works or benchmark datasets?
- Please provide an answer for the question you ask in section 4.3, even if that answer is "there is insufficient data to answer this question".
- For Figure 4C, we see performance on GSM8k decrease for most runs wrt the base model, for both full finetuning and LoRA. Does this indicate a mismatch between the GSM8k and WebMath dataset? It would be useful to provide more intuition for the lack of learning here as this figure seems to be the odd one out.
- There are many knobs to tune for LLM finetuning, especially when switching from one domain to another. How were these parameters chosen? It would be beneficial to explain the training procedure in more detail if possible, including what is done with tokens+embeddings (if anything).
- Related: Is the tokenizer frozen for code and math finetuning runs? In the domain shift setting, would a model benefit from adding new tokens or adapting the tokenizer, potentially reaching better performance or preventing forgetting if tokens for code/maths are added? See e.g. "Efficient domain adaptation of language models via adaptive tokenization", Sachidananda 2021. Some discussion on why these parameters are good under domain shift would strengthen the claims, IMO.

**Strengths And Weaknesses:**

Strengths:
- The work investigates a large number of settings and benchmarks: continued pretraining vs instruction finetuning, benchmarks for Python code and maths generation, using multiple LoRA configurations per setting.
- The authors attempt to understand why full finetuning learns more by investigating regularization effect of LoRA and looking at the rank of weight matrices post-training. Although the observation that "less flexible models change less" may seem trivial, this investigation provides additional insight into this specific finetuning method, and supporting evidence that LoRA finetuning is not always a good idea or easy drop-in replacement for full finetuning.
- The authors provide practical recommendations for LoRA based finetuning (for IFT vs CPT), making this paper potentially useful reference material for practitioners - even if the used datasets are perhaps less common than some text datasets.

Weaknesses:
- The authors ask an important question in section 4.3: "The nontrivial question is: do LoRA and full finetuning differ in how they tradeoff learning and forgetting? Can LoRA achieve similar target domain performance but with diminished forgetting?". I don't believe that this specific question is answered in the paper as there is insufficient data to answer positively or negatively.
	- For code and math, it does like like we should use full finetuning (from section 4.1).
	- But: the authors do observe that there are outliers where LoRA performs as well while forgetting less. As far as I can tell, they do not draw conclusions from these observations however, nor make recommendations for when to use LoRA.
	- This work would have been stronger if full finetuning runs were run until they reach similar forgetting-or-learning performance, to enable direct comparison. At the moment the LoRA and full finetuning runs mostly live in different parts of the tradeoff curve. Of course, full finetuning is expensive so it is understandable that there are fewer points for this setup.

- The authors show results for multiple ranks in the appendix, but as far as I can tell only for a single value for alpha for some Code runs, while keeping it to 2*rank for the maths runs. Given that there are many setup differences between code and maths runs, including LoRA hyperparameters, it may be less meaningful to compare LoRA performance between these two settings. That said, I'm not sure if using the same LoRA configuration in both domains is useful either.
- The insights about forgetting and learning seem to depend strongly on the domain and dataset. The paper title makes the claim more general than it may be in practice. LoRA is often used in domains not related to maths and code generation, and we cannot say whether the same insights will hold there. Most LORA variations seem to look at GLUE or other text datasets. "Forgetting" is more difficult to test when going from text to text domains, but learning can probably be measured!

I don't think restricting to "just" Llama-2 is a downside of this study, this is a large model that's expensive to train/finetune, and the authors try two very large versions. Obviously the insights would be more general if other models were tried as well but I don't think that's a prerequisite for acceptance.

Additionally:
- I think the claims in the submission are accurate and convincing. My only (small) concern is that the general claim in the title has a caveat "in code and maths generation tasks". The authors provide clear evidence that LoRA learns less and forgets less, characterized by the learning/forgetting tradeoff curve. The study seems broad, looking at multiple datasets in domains in detail, even if it only tackles math and code (and not say, text generation in a specific domain). The authors make no claims that LoRA is as good as full finetuning, as only a handful of datapoints show this.
- I believe this study is interesting for TMLRs audience, especially the learning/forgetting tradeoff and the practical recommendations for LoRA based finetuning on small domains. Choosing whether to use LoRA or not based on specific points on the tradeoff curve could be useful in practice. Additionally, this study is one of the first to show that LoRA is not always a good drop-in replacement for full finetuning on specific domains.

---

> ### Author Response · Authors · 2024-06-19
> **Thank you for your thoughtful review!**
>
> We thank the reviewer for the thorough assessment of our work.
>
> > The authors ask an important question in section 4.3: "The nontrivial question is: do LoRA and full finetuning differ in how they tradeoff learning and forgetting? Can LoRA achieve similar target domain performance but with diminished forgetting?". I don't believe that this specific question is answered in the paper as there is insufficient data to answer positively or negatively. Requested change: please provide an answer for the question you ask in section 4.3, even if that answer is "there is insufficient data to answer this question".
>
> Section 4.3 will state that our results cannot conclusively tell whether LoRA and full finetuning live on same or different tradeoff curves. It will also discuss potential avenues of future work given these results.
>
> > This work would have been stronger if full finetuning runs were run until they reach similar forgetting-or-learning performance, to enable direct comparison. At the moment the LoRA and full finetuning runs mostly live in different parts of the tradeoff curve. Of course, full finetuning is expensive so it is understandable that there are fewer points for this setup.
>
> The revision will include full finetuning experiments for shorter durations. In response to your question below, we are currently running additional LoRA experiments with identical scaling factors across math and code (alpha=2r). We are hopeful that our new results will show increased overlap between full finetuning and LorA at least for high ranks.
>
> > The authors show results for multiple ranks in the appendix, but as far as I can tell only for a single value for alpha for some Code runs, while keeping it to 2*rank for the maths runs. Given that there are many setup differences between code and maths runs, including LoRA hyperparameters, it may be less meaningful to compare LoRA performance between these two settings. That said, I'm not sure if using the same LoRA configuration in both domains is useful either.
>
> We agree that a cleaner comparison of math and code should use the same scaling parameter alpha, and therefore, we are currently rerunning all experiments with alpha=2r, which is our biggest undertaking in this revision.
>
> > The insights about forgetting and learning seem to depend strongly on the domain and dataset. The paper title makes the claim more general than it may be in practice. LoRA is often used in domains not related to maths and code generation, and we cannot say whether the same insights will hold there. Most LORA variations seem to look at GLUE or other text datasets. "Forgetting" is more difficult to test when going from text to text domains, but learning can probably be measured!
>
> We agree: questions of learning and forgetting depend on domain and dataset. While we ran extensive experiments on two domains and four datasets, we cannot make statements on domains which we have not tested, such as standard text datasets. We will acknowledge this in the updated manuscript.
> A handful of papers have established that LoRA is competitive with full finetuning for certain tasks/datasets/model sizes (e.g. RoBERTa on GLUE). These works focus on substantially smaller Encoder-style models and we do not disagree with their claims. We chose instead to test the differences between the two techniques in larger Decoder-style models and verifiable tasks.
>
> > Please provide more information about the experimental setup, especially the motivation for using different LoRA configurations between maths and code settings. Are hyperparameter choices motivated by the original works or benchmark datasets?
>
> We will provide more information about the experimental setup. As noted above, we are currently working to equalise the LoRA configurations for code and math. Our LoRA hyperparameter choices were not motivated by the original works – as these reported full finetuning results – but rather our preliminary hyperparameter sweeps.
>
> > For Figure 4C, we see performance on GSM8k decrease for most runs wrt the base model, for both full finetuning and LoRA. Does this indicate a mismatch between the GSM8k and WebMath dataset? It would be useful to provide more intuition for the lack of learning here as this figure seems to be the odd one out.
>
> We are currently rerunning the experiments in Fig. 4C with more tokens, to get closer to the regime in the Starcoder experiment. We suspect that CPT with <1B tokens (which is unusually low) might lead to degradation in evaluation because of the fast ramp up during learning rate warmup. As the reviewer suggested, the grade-school level questions in GSM8K might also not capture the full breadth of advanced mathematics in OpenWebMath, which we will discuss.
>
> (Continued)

---

> > ### Author Response · Authors · 2024-06-19
> > **Continued response**
> >
> > > There are many knobs to tune for LLM finetuning, especially when switching from one domain to another. How were these parameters chosen? It would be beneficial to explain the training procedure in more detail if possible, including what is done with tokens+embeddings (if anything).
> >
> > We will include a more detailed experimental section describing our training and evaluation protocols. As is common practice for LoRA and QLoRA methods, the input and output embedding layers are not trained. Recent open-source work argues that it might be beneficial to supplement LoRA with full-finetuning of these two input and output layers (additional ~200M parameters for a 7B model). We certainly see the value of this approach, but we view it as a hybrid of LoRA and full finetuning, and have therefore postponed our investigation of it. Moreover, this hybrid approach involves further hyperparameter optimization: the input and output layers require tuning their own separate learning rates (training becomes unstable if using the LoRA learning rate for these layers well). We will explicitly address these input and output embedding layers in the aforementioned experimental section.
> >
> > > Related: Is the tokenizer frozen for code and math finetuning runs? In the domain shift setting, would a model benefit from adding new tokens or adapting the tokenizer, potentially reaching better performance or preventing forgetting if tokens for code/maths are added? See e.g. "Efficient domain adaptation of language models via adaptive tokenization", Sachidananda 2021. Some discussion on why these parameters are good under domain shift would strengthen the claims, IMO.
> >
> > We agree that the downstream implications of tokenization strategies are an important area for future research. And we will explain our tokenizer choices in the revised experimental design section. In LLM continual learning, the tokenizer commonly remains fixed. The Llama tokenizer is already designed for code and math, and indeed, it is used by both the CodeLlama [1] and Llemma (math; [2]) models. Additionally, we do not see how changing the tokenizer would help in mitigating forgetting.
> >
> > We hope that our comments address the reviewer's feedback.
> >
> > [1] Code Llama: Open Foundation Models for Code, Roziere et al., 2023
> > [2] Llemma: An Open Language Model For Mathematics, Azerbayev et al., 2024

---

> > ### Comment · Reviewer_eciD · 2024-06-25
> > **Thanks for the response**
> >
> > This response addresses most of my concerns, especially if more points will be added to the forgetting-learning tradeoff curve. Even if this cannot the answer from 4.3 in full, it may be a bit more conclusive that LoRA can produce a better tradeoff. Thanks for running additional experiments, I do believe these will make the paper stronger.
> >
> > I think it's alright to make broader statements if the data supports it, but discussing some of the caveats or limitations of the study would indeed be good. The choice for LoRA should depend on the setting.
> >
> > My comment on tokenizer finetuning is indeed not so relevant here, and it's also not how LoRA is typically used.
> >
> > With the proposed changes I believe this paper will be a useful contribution, interesting to many readers. Thanks!

---

### Review · Reviewer_dMFP · 2024-06-01

**Summary Of Contributions:**

This paper conducts a comprehensive analysis on LoRA fine-tuning and full fine-tuning. Under various fine-tuning tasks, the authors make several interesting observations. On the downstream tasks, LoRA underperforms full fine-tuning. However, LoRA incurs less loss of the performance on pretraining tasks. Also, LoRA achieves stronger regularization compared to common techniques. The authors also analyzed the rank of perturbations from full fine-tuning and observed that they are much larger than the rank number used in LoRA, possibly explaining the gap between LoRA and full-finetuning.

**Audience:**

Yes

**Broader Impact Concerns:**

I do not have concerns regarding the broader impacts of this paper.

**Claims And Evidence:**

Yes

**Requested Changes:**

See weaknesses for details. For me the biggest concern is that this paper lacks a clear discussion on the existing studies comparing LoRA and full fine-tuning, e.g. performance and efficiency. I would suggest the authors describe the existing consensus regarding this tradeoff clearly in the introduction. If there is no study on this before, it should also be mentioned.

Also, the authors should clarify the meaning of the term “regularization” used in the paper. I think the current description is too vague and short.

**Strengths And Weaknesses:**

Strengths:
1. The paper is well-written and the key message is presented clearly, supported by strong empirical evidence.

2. This paper conducts a deep analysis on LoRA and full-finetuning. One notable thing is that the authors have evaluated on many code and math datasets.

3. The question that this paper tries to answer is important in practice. With the huge computational resources required to train LLMs, we often faces the problem as to what to choose between LoRA and full fine-tuning. I think this paper provides a valuable contribution on this front.


Weaknesses:
1. The authors have found that section 4.1 LoRA underperforms full fine-tuning on math and coding tasks. This seems to be expected, given that LoRA uses much less parameters. I wonder if this finding is novel or just verifying existing hypothesis?

2. I think this paper lacks some discussion on the existing practice of fine-tuning on the match and code datasets. For example, one question would be when people fine-tune on these math or code datasets, what fine-tuning strategies do they often use? Are there tradeoffs that people have observed before or is this the first work?

3. There is some ambiguity around the regularization here. In section 4.4, the authors mentioned that “we define regularization (loosely) as a training mechanism that keeps the finetuned LLM similar to the base LLM”. However, this does not seem to be the common meaning of regularization, which I think refers to techniques that helps the training trajectory easier. So my question is that why would people care about the regularization property as defined in the paper? It would also be good to add more clarification on what the authors mean by regularization in the paper.

4. Beyond code and math tasks, I think another important and common fine-tuning direction is fine-tuning for chatting purposes. I wonder if the authors have done experiments on this.

5. The paper concluded with some suggestions on practical takeaways for LoRA fine-tuning, i.e. Section 4.6. However, I find this part unrelated to the topic of this paper, which is to study the tradeoff between LoRA and full-finetuning. I would suggest the paper add more aspects on how practitioners can better choose between LoRA and full fine-tuning, given the observations from this paper.

---

> ### Author Response · Authors · 2024-06-19
> **Thanks for your review**
>
> We are thankful for the constructive assessment of our work and agree with all of your suggestions. Namely:
> > The authors have found that section 4.1 LoRA underperforms full fine-tuning on math and coding tasks. This seems to be expected, given that LoRA uses much less parameters. I wonder if this finding is novel or just verifying existing hypothesis?
>
> We are glad that the reviewer sees our findings as intuitive. However, the two major methods papers on the topic, LoRA [1] and QLoRA [2], report that LoRA performs better or equivalent to full finetuning, respectively. A recent analysis [3] makes a similar point; and this sentiment is echoed in an array of industry blog posts as well (e.g. [4]). At the same time, there is scarce work showing – implicitly or explicitly – that LoRA underperforms full finetuning (e.g. [5,6]). We believe our experimental setup is a useful datapoint in this space since it compares full finetuning and LoRA on several data domains and training regimes, performing careful hyperparameter searches, and with meaningful downstream evaluations.
>
> > I think this paper lacks some discussion on the existing practice of fine-tuning on the match and code datasets. For example, one question would be when people fine-tune on these math or code datasets, what fine-tuning strategies do they often use? Are there tradeoffs that people have observed before or is this the first work?
>
> We agree with the reviewer, and our revised Related Work section will review previous work studying continual learning on math and code datasets (via continued pretraining or instruction finetuning). Our work, we believe, covers the majority of training settings used for code and math.
>
> Most previous work on the topic aims to introduce new datasets on which models can be trained to achieve state of the art. PEFT techniques, like LoRA, are typically used only as a necessity when getting to bigger models (e.g., [7]). Few works study the efficacy of PEFT methods on these domains, one of them is [6], which compared PEFT vs full finetuning for code, reporting that full finetuning led to better downstream performance. In summary, whereas existing work revolves around _expanding LLM capabilities_ in the math and code domains, we used this setup as a testbed for comparing LoRA versus full finetuning.
>
> > There is some ambiguity around the regularization here. In section 4.4, the authors mentioned that “we define regularization (loosely) as a training mechanism that keeps the finetuned LLM similar to the base LLM”. However, this does not seem to be the common meaning of regularization, which I think refers to techniques that helps the training trajectory easier. So my question is that why would people care about the regularization property as defined in the paper? It would also be good to add more clarification on what the authors mean by regularization in the paper.
>
> This point is well taken. We will avoid the use of “regularization”, in favor of terminology that is more explicit about learning and forgetting in downstream tasks.
>
> > Beyond code and math tasks, I think another important and common fine-tuning direction is fine-tuning for chatting purposes. I wonder if the authors have done experiments on this.
>
> In response to the reviewer’s comment, we are running a new experiment on the Tülu-2 finetuning dataset [5], evaluating chat-quality with MT-Bench.
>
> > The paper concluded with some suggestions on practical takeaways for LoRA fine-tuning, i.e. Section 4.6. However, I find this part unrelated to the topic of this paper, which is to study the tradeoff between LoRA and full-finetuning. I would suggest the paper add more aspects on how practitioners can better choose between LoRA and full fine-tuning, given the observations from this paper.
>
> We agree. The revised version will better streamline the connection between the core experiments and our hyperparameter sensitivity analyses and recommendations.
>
> > Requested change: For me the biggest concern is that this paper lacks a clear discussion on the existing studies comparing LoRA and full fine-tuning, e.g. performance and efficiency. I would suggest the authors describe the existing consensus regarding this tradeoff clearly in the introduction. If there is no study on this before, it should also be mentioned.
>
> The introduction will review work comparing LoRA vs full finetuning, as detailed above. It will make the point that the jury is still out: a number of methodological papers show that LoRA is on par with full finetuning, whereas some large-scale empirical investigations reach a different conclusion. We will mention potential differences between code/math vs text domains.
>
> [1] Hu et al., 2021
>
> [2] Dettmers et al., 2023
>
> [3] “A Closer Look at the Limitations of Instruction Tuning” (Ghosh et al., 2024)
>
> [4] https://sebastianraschka.com/blog/2023/llm-finetuning-lora.htm
>
> [5] Tulu 2, Ivison et al. 2023
>
> [6] Astraios, Zhuo et al., 2024
>
> [7]  MetaMathQA, Yu, Jiang, et al., 2023

---

> > ### Comment · Reviewer_dMFP · 2024-06-23
> > **Thanks for the response!**
> >
> > Thanks for the detailed response. My concern is mostly addressed and I hope that the revised manuscript reflects those changes. Given the widespread use of fine-tuning LLMs in practice, I believe this paper is a valuable contribution that is of great interest to the research community.

---

### Review · Reviewer_vfzD · 2024-06-08

**Summary Of Contributions:**

The authors provide an in-depth analysis of a popular parameter-efficient fine-tuning method, LoRA.
They compare LoRA with full fine-tuning in both math and coding domains and find that LoRA tends to forget
less the source domain, while learning less than full fine-tuning, which is more sample-efficient.
They also show that LoRA is a regularizer that promotes generation diversity.
Another interesting finding is that, contrary to popular belief, full-parameter fine-tuning finds high rank weight
perturbations, which shakes one of the key motivations of utilizing LoRA.

**Audience:**

Yes

**Claims And Evidence:**

Yes

**Requested Changes:**

- Please include missing inference hyperparameters used for evaluation in the paper.
- What is the impact on pass@k; k>1 for LoRA compared to full fine-tuning? Given the claim that LoRA promotes generation diversity,
  it would be interesting to see how this affects the pass@k metric with ks greater than 1.
  For this experiment, the authors should consider using a higher temperature parameter than 0.2.
  [1] finds a temperature of 0.8 to be optimal for this task.
  One way to show this is to graph the pass@k metric against k for both LoRA and full fine-tuning.

[1] Chen et al., "Evaluating Large Language Models Trained on Code"

**Strengths And Weaknesses:**

### Strengths

- A large number of useful contributions are made throughout this paper. This is a very insightful paper that will be of great interest to the community.
- The experimental setup is sound and the results are well presented. The authors fine-tune the models several times to reduce the variance in the results,
  which is a good practice and will help to ensure the results are reliable. Additionally, the authors utilize open-access datasets and models, which will help
  in the reproducibility of the results and foster further research in this area.
- The hyper-parameter analysis for LoRA is a great addition. It's widely known that LoRA is sensitive to hyper-parameters, and the authors provide a
  comprehensive analysis of the hyper-parameters that affect LoRA the most.
  Interestingly, they find that selecting appropriate target modules is more important than rank itself. Future work may focus on understanding why this is the case.

### Weaknesses

- While the authors provide inference hyperparameters for HumanEval and GSM8K, they do not provide them for HellaSwag, Arc, and WinoGrande.
  This information is vital for reproducibility and should be included in the paper.
  Additionally, for HumanEval and GSM8K, the authors should provide top-p and top-k hyperparameters used, if any.
- The analysis on sample diversity for LoRA could be improved. The authors utilize string match to measure diversity, which is a very limited metric,
  especially for code generation tasks.

---

> ### Author Response · Authors · 2024-06-19
> **Thank you.**
>
> Thank you for your positive assessment of our work, and for the useful suggestions!
>
> > Weakness: While the authors provide inference hyperparameters for HumanEval and GSM8K, they do not provide them for HellaSwag, Arc, and WinoGrande. This information is vital for reproducibility and should be included in the paper. Additionally, for HumanEval and GSM8K, the authors should provide top-p and top-k hyperparameters used, if any.
> Requested change: Please include missing inference hyperparameters used for evaluation in the paper.
>
> The revised manuscript will further detail the evaluation hyperparameters we used, and also link to the evaluation repositories we used, to ensure reproducibility at this stage as well. For HumanEval and GSM8K, we followed common practices – set out in BigCode Leaderboard – and used temp=0.2 and top_p=0.95 for 0-shot HumanEval and temperature=0 and 5-shot for GSM8k. As for HellaSwag, ARC-challenge, and WinoGrande, we will clarify that these involve multiple-choice questions that use the predicted logits for calculating accuracy, and thus don’t have generation hyperparameters.
>
> > The analysis on sample diversity for LoRA could be improved. The authors utilize string match to measure diversity, which is a very limited metric, especially for code generation tasks.
>
> We agree that strict string matching is not a sensitive metric of diversity. Nevertheless, this metric sufficed to showcase that full finetuning remarkably generates a large fraction of strictly identical generations, compared to LoRA and the base model. More flexible string matching is indeed preferred, and an interesting direction for future work.
>
> > Requested change: What is the impact on pass@k; k>1 for LoRA compared to full fine-tuning? Given the claim that LoRA promotes generation diversity, it would be interesting to see how this affects the pass@k metric with ks greater than 1. For this experiment, the authors should consider using a higher temperature parameter than 0.2. [1] finds a temperature of 0.8 to be optimal for this task. One way to show this is to graph the pass@k metric against k for both LoRA and full fine-tuning.
>
> We have run the new analysis suggested by the reviewer, using temperature=0.8 and computing HumanEval pass@k for k=1, …, 256 for both LoRA (at different ranks, targeting all modules) and full finetuning. See the attached figure `pass-at-k-up-to-256.png` in the Supplementary Material, which shows that:
> * For both LoRA and full finetuning, pass@k increases as k increases.
> * The relative ordering of Full Finetuning > LoRA rank 256 > rank 64 > rank 16 is preserved as k increases.
>
> We will include this figure in the revised appendix. It is worth mentioning that even though LoRA does worse than full finetuning for a 7B model, it does quite well at a relatively difficult task as k increases. Even with r=16, 80% accuracy at pass@64 is surprisingly good. This is reminiscent of the trend in Figure 5 of the original HumanEval paper [1], where higher temperatures lead to improved accuracy with increasing k.
>
> Thank you again for this suggestion which will make a very interesting addition to our paper.
>
> [1] “Evaluating Large Language Models Trained on Code”, Chen et al. 2021. https://arxiv.org/pdf/2107.03374v2

---

> > ### Comment · Reviewer_vfzD · 2024-06-23
> > **Thanks**
> >
> > I thank the authors for their response. The response has covered all of my concerns. I believe this work to be highly insightful and a valuable contribution to the scientific community.

---

### Author Response · Authors · 2024-08-13
**Camera-ready version submitted**

We thank the editor and reviewers for their valuable feedback. We have addressed all the comments and have run further experiments and analyses accordingly. We believe that the paper is greatly improved and hope that it will be a valuable resource to the growing research community interested in LLM finetuning.

---

### Decision · Action_Editor_KgcR · 2024-07-12

**Recommendation:** Accept with minor revision

**Comment:**

This paper conducts a comprehensive analysis on LoRA fine-tuning and full fine-tuning. Under code and math tasks, the authors empirically observed that LoRA learns less and forgets less. After author rebuttal, it received Accept, Accept, and Leaning Accept recommendations.

All the reviewers agree that (1) the paper is well-written and the key message is presented clearly, supported by strong empirical evidence, (2) the question that this paper tries to answer is important in practice, and the paper is insightful and will be of great interest to the community.

On the other hand, reviewers have asked many questions, and the authors have also promised many changes. However, the current draft has not reflected these. Therefore, the AC would like to recommend accept, depending on whether the promised changes have been made.

The authors have promised adding additional new results, including:

1. the new analysis suggested by the reviewer, using temperature=0.8 and computing HumanEval pass@k for k=1, …, 256 for both LoRA (at different ranks, targeting all modules) and full finetuning.

2. the new experiment results on the Tülu-2 finetuning dataset, evaluating chat-quality with MT-Bench.

3. full finetuning experiments for shorter durations; we are currently running additional LoRA experiments with identical scaling factors across math and code (alpha=2r).

4. rerunning the experiments in Fig. 4C with more tokens, to get closer to the regime in the Starcoder experiment.

In terms of paper writing, the authors have promised to include the following revision:

1. detail the experimental setup and evaluation hyper-parameters used, and also link to the evaluation repositories we used, to ensure reproducibility.

2. revise Related Work section to review previous work studying continual learning on math and code datasets (via continued pretraining or instruction finetuning).

3. try to avoid the use of “regularization”, in favor of terminology that is more explicit about learning and forgetting in downstream tasks.

4. better streamline the connection between the core experiments and our hyperparameter sensitivity analyses and recommendations.

5. the introduction will review work comparing LoRA vs full finetuning, and make the point that the jury is still out.

6. Section 4.3 will state that our results cannot conclusively tell whether LoRA and full finetuning live on same or different tradeoff curves. It will also discuss potential avenues of future work given these results.

**Audience:**

Yes

**Claims And Evidence:**

Yes